# Understanding Robust Learning through the Lens of Representation Similarities

**Christian Cianfarani**[*]
Department of Computer Science
University of Chicago

**Arjun Nitin Bhagoji**[*]
Department of Computer Science
University of Chicago

**Vikash Sehwag**[*]
Department of ECE
Princeton University

**Ben Zhao**
Department of Computer Science
University of Chicago

**Haitao Zheng**
Department of Computer Science
University of Chicago

**Prateek Mittal**
Department of ECE
Princeton University

## Abstract

Representation learning, *i.e.* the generation of representations useful for downstream applications, is a task of fundamental importance that underlies much of the success of deep neural networks (DNNs). Recently, *robustness to adversarial examples* has emerged as a desirable property for DNNs, spurring the development of robust training methods that account for adversarial examples. In this paper, we aim to understand how the properties of representations learned by robust training differ from those obtained from standard, non-robust training. This is critical to diagnosing numerous salient pitfalls in robust networks, such as, degradation of performance on benign inputs, poor generalization of robustness, and increase in over-fitting. We utilize a powerful set of tools known as representation similarity metrics, across three vision datasets, to obtain layer-wise comparisons between robust and non-robust DNNs with different training procedures, architectural parameters and adversarial constraints. Our experiments highlight hitherto unseen properties of robust representations that we posit underlie the behavioral differences of robust networks. We discover a lack of specialization in robust networks' representations along with a disappearance of 'block structure'. We also find overfitting during robust training largely impacts deeper layers. These, along with other findings, suggest ways forward for the design and training of better robust networks.

## 1 Introduction

Representation learning is fundamental to machine learning and a task on which deep neural networks perform remarkably well. Given complex high-dimensional input signals, such as images, speech, or text, deep neural networks (DNNs) can learn meaningful, lower-dimensional representations [33, 6, 57, 13, 24, 12, 66]. The downstream utility of these representations underlies DNNs' success on different tasks [5, 15, 51, 36, 3]. For standard image classification, neural networks learn well-separated representations of inputs from different classes to achieve high accuracy. The discovery of adversarial examples [4, 54], perturbed inputs that induce high error in model predictions (even for well-trained classifiers) has motivated the need for another desirable property: *adversarial robustness*. Robust training methods typically protect against adversarial examples during the training phase by converting the standard empirical risk minimization objective to a min-max one (Figure 1) [35, 46, 65]. However, they typically have far worse performance on benign inputs, converge

---

[*]Equal Contribution

36th Conference on Neural Information Processing Systems (NeurIPS 2022).

slower and appear to require larger architectures to be effective [17, 38, 56, 60]. Given the importance of representation learning for good downstream performance, we posit that an effective method to identify potential improvements for robust training is via developing a better understanding of robustly learned representations [2].

Our approach is to juxtapose representations of non-robust and robust networks and analyze them through a unified lens. Aggregate metrics like loss and accuracy do shed some light on the differences, but these are task-driven and do not admit a layer-wise comparison of representations [1, 20]. We need a comprehensive analysis of the impact of choice of architectural parameters like depth and width, nature of the threat model and corresponding attack strength, and the choice of dataset on robustly learned representations and their link to downstream performance.

To this end, we leverage recent work on representation similiarity (RS) that enables fine-grained comparisons of representations obtained across all layers in a deep neural network (DNN) [31, 44]. Our key findings, using both benign and adversarial inputs as probes over CIFAR-10 [32] and the Imagenette and Imagewoof [25] subsets of the Imagenet [10] dataset, are:

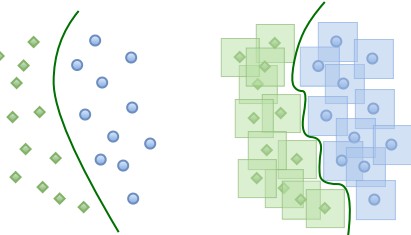

Non-robust classification   Robust classification

Figure 1: **Non-robust vs robust training.** Adversarially robust training aims to learn decision boundaries that are robust to adversarial perturbations ($\ell_\infty$ in this sketch), naturally learning more complex decision boundaries than non-robust training.

**Robust representations are less specialized (§3):** Robust networks exhibit much higher similarity across spatially distant layers compared to non-robust networks, which have far more localized similarity. The latter thus exhibit the expected specialization in layers, which is unexpectedly missing in the former. Further, we find reducing the width of the network has a similar effect to increasing the budget used in robust training, with both changes increasing long-range similarity among the learned representations. This indicates that the relative capacity of a network drops as the budget is increased.

**Early layers are largely unaffected by adversarial examples (§4):** The representations of perturbed inputs from non-robust and robust networks are, as expected, starkly different. On the other hand, the representations of benign and perturbed inputs from robust networks are indistinguishable from one another with regards to representation similarity metrics.

**Deeper layers overfit (§5):** For both non-robust and robust training, early layers in the network converge within the first few epochs, and exhibit very little variation in subsequent training. This indicates they perform similar roles in both benign and robust networks. Deeper layers converge slower while visiting drastically different representations over the course of training. These different representations usually occur when the loss indicates overfitting, leading us to conclude that *deeper layers overfit during robust training*.

**Visually dissimilar adversarial constraints can lead to similar robust representations (§6):** Regardless of the adversarial constraint, we find a high degree of similarity between the representations of perturbed and benign inputs for robust networks. Representations from networks robust to different, but allied (*e.g.* $\ell_p$), threat models are similar across attack strength. We find surprising similarities between the representations of visually unrelated threat models at lower attack strengths.

**What do our findings imply?** The lack of specialization in robust networks even with heavily overparametrized architectures suggests a need for explicit tools to differentiate between layers during training. Further, the overfitting we find in deeper layers of neural networks is an indication that layer-wise specialized regularization methods need to be adopted for robust training. We hope the lessons from this paper and our accompanying website [3] with open-sourced code spur the development of robust learning-specific architectures and training methods.

**Paper Layout.** We provide an overview of RS tools and robust training methods in §2. We compare representations from non-robust and robust networks using benign inputs in §3. §4 repeats this

---

[2]We note that we use the word 'robust' as shorthand for 'adversarially robust' throughout the paper. When considering other types of robustness, such as to common corruptions, we explicitly mention it.

[3]https://robustrs.github.io/

| Network | Benign accuracy | | | Robust accuracy | | |
| --- | --- | --- | --- | --- | --- | --- |
| | Train | Test | $\Delta$ | Train | Test | $\Delta$ |
| Non-robust | 100.0% | 94.9% | 5.1% | 0.0% | 0.0% | 0.0% |
| Robust | 100.0% | 87.6% | 12.4% | 98.2% | 44.2% | 54.0% |

Table 1: Performance of non-robust and robust Wide Resnet 28-10 (WRN-28-10) on the CIFAR-10 dataset. Note, in particular, the high generalization gap in robust networks .

analysis with perturbed inputs. The evolution of robust representations over training is studied in §5 while the impact of different threat models is in §6. §7 discusses lessons and limitations.

## 2 Background and Setup

We provide a brief overview of the techniques we use to train robust networks and the tools we use to probe the intermediate representations of neural networks. Further background and related work is in Appendix A.

### 2.1 Adversarial Examples and Robust Learning

Adversarial examples are perturbed inputs to machine learning models that intentionally induce misclassification [4, 54]. For an input $\mathbf{x}$, an untargeted adversarial example with respect to a model $f$ and loss function $\ell(\cdot)$ is generated by adding a perturbation $\boldsymbol{\delta}$ such that $\boldsymbol{\delta} = \arg\max_{\mathbf{x}+\boldsymbol{\delta} \in N(\mathbf{x})} \ell(f(\mathbf{x}+ \boldsymbol{\delta}), y)$, where $y$ is the label of the original class and $N(\cdot)$ represents the adversarial constraint or neighborhood. The most commonly used constraints are $L_p$-ball based, although we also experiment with other unrelated classes of attacks, such as JPEG, Gabor, or Snow based perturbations [28]. We characterize activations of internal layer in a network as: 1) *Benign representation:* if input is non-adversarial 2) *Adversarial representation:* if input is an adversarial example.

Adversarial training [35] has emerged as the most effective methods to train robust classifiers [7], where it accounts for adversarial examples during training. It modifies the empirical risk minimization procedure in non-robust training to a min-max optimization, where the inner maximization focuses on finding the strongest adversarial example for each input during training:

$$\min_{\theta} \mathbb{E}_{(\mathbf{x},y) \in \mathcal{D}} \left[ \max_{\mathbf{x}+\boldsymbol{\delta} \in N(\mathbf{x})} \ell(f_{\theta}(\mathbf{x} + \boldsymbol{\delta}), y) \right], \tag{1}$$

where $\mathcal{D}$ is a data distribution and $f_{\theta}$ is a model with parameters $\theta$. Newer methods, like TRADES [65], attempt to improve on adversarial training by better balancing the tradeoff between robustness and accuracy. We refer to models trained using methods that account for adversarial examples as *robust* models and those trained without these methods as *non-robust* models.

We refer to accuracy on non-perturbed input as *benign accuracy* and on adversarial examples as *robust accuracy*. Adversarial training is well known to trade-off benign accuracy to gain robustness and exhibits a high generalization gap in robust accuracy (as we show in Table 1 for the CIFAR-10 dataset).

### 2.2 Probing Neural Network Representations

Representation similarity (RS) metrics allow for the quantitative comparison of the activations of different sets of neurons on a given dataset. Comparing learned, internal representations within and between neural networks of different architectures is important for understanding the performance and behavior of these networks under changing conditions. Several metrics have been proposed for this purpose: Canonical Correlation Analysis (CCA) [22] and its extensions, Singular Vector CCA (SVCCA) [44] and Projection Weighted CCA (PWCCA) [37], Centered Kernel Alignment (CKA) [31], the Orthogonal Procrustes distance [14] and model stitching [2, 9]. These metrics can all handle different sized representations, but require that the number of datapoints be the same. Other work has used the fitting of sparse linear layers to understand the behavior of deeper layer representations [59, 47].

We choose CKA to measure representation similarity, due to its desirable properties such as orthogonal and isotropic invariance, and wide use in previous work [40, 42, 43, 58]. We also find CKA to be computationally more efficient ($10\times$ and $30\times$ faster than Procrustes and SVCCA respectively), while yielding similar high-level insights (see Appendix B.1). Ding et al. [14] have shown that CKA is insensitive to deletions of principal components of representations that strongly correlate with model accuracy. However, since we do not compare neural networks with representations of reduced dimensions, this does not affect our results.

**Centred Kernel Alignment (CKA):** Let $\boldsymbol{X} \in \mathbb{R}^{n \times p_1}$ with rows $\boldsymbol{x}_i$, and $\boldsymbol{Y} \in \mathbb{R}^{n \times p_2}$ with rows $\boldsymbol{y}_i$ be the activation matrices of layers with $p_1$ and $p_2$ neurons respectively, defined on the same set of $n$ points. These activation matrices are then the representations induced by a particular layer. We also define a $n \times n$ centering matrix $\boldsymbol{H} = \boldsymbol{I}_n - \boldsymbol{1}\boldsymbol{1}^\top$. The CKA between these activation matrices is then

$$\mathrm{CKA}(\boldsymbol{K}, \boldsymbol{L}) = \frac{\mathrm{tr}(\boldsymbol{K}\boldsymbol{H}\boldsymbol{L}\boldsymbol{H})}{\sqrt{\mathrm{tr}(\boldsymbol{K}\boldsymbol{H}\boldsymbol{K}\boldsymbol{H})\mathrm{tr}(\boldsymbol{L}\boldsymbol{H}\boldsymbol{L}\boldsymbol{H})}}, \tag{2}$$

where $\boldsymbol{K}_{ij} = k(\boldsymbol{x}_i, \boldsymbol{x}_j)$, $\boldsymbol{L}_{ij} = l(\boldsymbol{y}_i, \boldsymbol{y}_j)$, with $k$ and $l$ being kernels. When the kernels are linear and the activation matrices are centered, we get the Linear CKA between activation matrices to be

$$\text{Linear CKA} = \frac{\|\boldsymbol{Y}^\top \boldsymbol{X}\|_F^2}{\|\boldsymbol{X}^\top \boldsymbol{X}\|_F \|\boldsymbol{Y}^\top \boldsymbol{Y}\|_F} = \frac{\langle \mathrm{vec}(\boldsymbol{X}\boldsymbol{X}^\top), \mathrm{vec}(\boldsymbol{Y}\boldsymbol{Y}^\top) \rangle}{\|\boldsymbol{X}^\top \boldsymbol{X}\|_F \|\boldsymbol{Y}^\top \boldsymbol{Y}\|_F}. \tag{3}$$

Since Kornblith et al. [31] find the use of a linear kernel to be as accurate as and much faster than the RBF kernel, we focus only on linear kernels. To reduce memory usage, we use online CKA which uses batching to get an unbiased estimate of CKA [42].

## 2.3 Experimental Setup

We provide a brief overview of our experimental setup here. All of our experiments were run on machines with either NVidia Titan RTX GPUs with 24GB of memory or RTX A4000 GPUs with 16GB of memory. Further details are in Appendix A.

**Models and datasets.** We consider three commonly used image datasets: CIFAR-10 [32], ImageNette [25], and ImageWoof [25], where the latter two are subsets of ImageNet [11], which we use due to the high cost of adversarial training on the full ImageNet dataset. We use the Wide Resnet [63] class of models on CIFAR-10 dataset and ImageNet-based ResNets for ImageNette and Imagewoof datasets.

**Adversarial training.** We follow standard convention with $\ell_\infty$ perturbations and use $\epsilon = \frac{8}{255}$ for CIFAR-10 and $\epsilon = \frac{4}{255}$ for ImageNet based datasets. We use 10-step projected gradient descent (PGD) attack during training and use 20 steps at test time to calculate adversarial representations.

**Representation similarity.** We will be using mainly using CKA to compare activation matrices derived from different benign or adversarial images. In online CKA, we use a batch-size of 1024 and take 3 passes over the dataset to reduce any stochasticity in the output similarity score. In layer-wise similarity analyses, we consider all layers, including convolutions, pooling, activation, and batch-normalization layers.

## 3 How do representations from robust and non-robust networks differ?

Robust training is expected to be a harder training objective than non-robust training, since it optimizes for both accuracy and robustness. The resulting trade-off is also reflected in final performance, where the benign accuracy of robust networks is much lower than non-robust networks [35]. In this section, we aim to understand the impact of robust training on internal representations, in particular by comparing benign representations in non-robust and robust networks. Further experiments and results from this section are included in Appendix B. We take the following two-fold approach:

**1) Comparing characteristics: Absence of block structure.** When comparing layerwise similarity in non-robust networks, numerous works have revealed a 'block structure', with high similarity between the layers comprising a block, and low similarity outside it [31, 42, 45] (Non-robust network plots in Figure 2).We observe that when moving from benign to robust training, while keeping

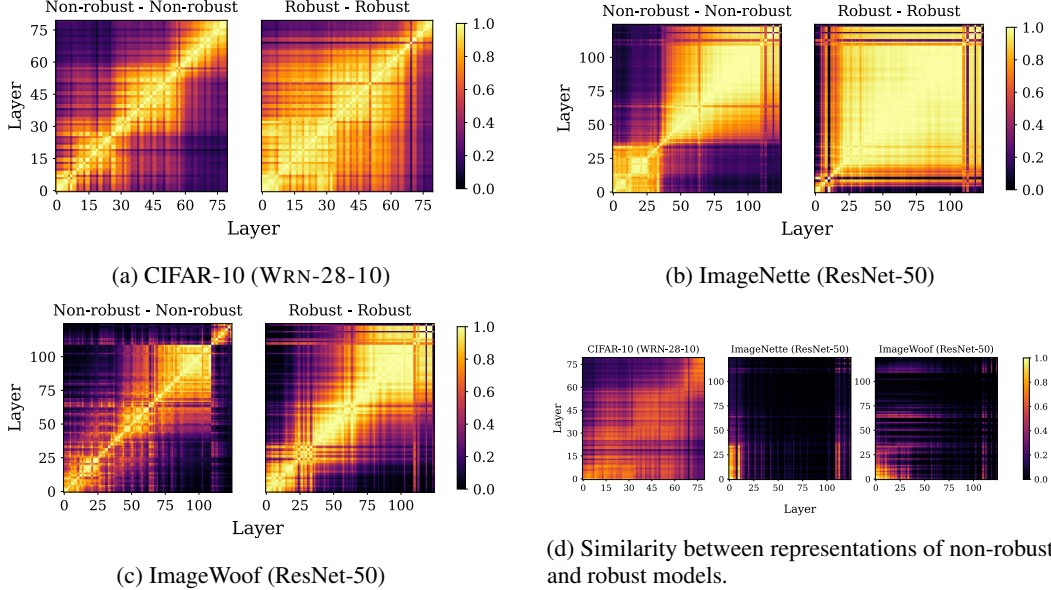

Figure 2: **Stark differences in the cross-layer similarity and block structure for robust networks.** We measure the similarity of *benign representations* for three datasets across all pairs of layers in both non-robust and robust networks. *(a, b, c):* Robust networks have an amplified degree of cross-layer similarity, even at long-range, which diminishes the well known block structure characteristic observed in non-robust networks [42]. *(d)* We observe low similarity between representations of non-robust and robust models with identical experimental setups.

everything else constant, the *block structure significantly fades*. The much weaker block structure in robustly trained networks is accompanied by the presence of *long-range similarities* between layers, with layers having high similarity to those much farther away in the network (Figure 2).

**2) Direct comparison: Poor similarity between non-robust and robust network representations.** To further investigate the unique characteristics of robust networks, we conduct a cross-layer comparison of benign representations in non-robust and robust networks (Figure 2d). We find a very low degree of similarity across both network representations, which further shows that robust networks had indeed learned strikingly different representations than non-robust networks.

We also show that both of these observations generalize across architectures, datasets and threat models (see Appendix B). We now delve deeper into *analyzing cross-layer characteristics of robust networks representations*.

**Impact of robust training strength.** We investigate this disappearance of block structure in robustly trained networks further by varying both the adversarial budget ($\epsilon$) used for training and the width of the network (Figure 3). As the training $\epsilon$ increases, the block structure becomes fainter, eventually disappearing entirely. The block structure is also impacted by network width, with it being marginally more pronounced in larger networks. However, this distinction is largely absent at larger values of $\epsilon$. This establishes that increasing the value of $\epsilon$ is making the task more complex, so maintaining the same model capacity leads to a decrease in relative capacity (indicated by high similarity across representations). This conclusion relates to several findings in the adversarial robustness literature on the topic of low-rank representations. Encouraging low dimensionality of learned representations through regularization has been found to increase adversarial robustness [49, 34, 39]. While it is outside the scope of this work, it would be interesting for future work to study if the task of learning low-dimensional robust representations is inherently more complex, leading to higher utilization of model capacity and thereby reducing differentiation between layers.

**Impact of architecture.** Previously Nguyen et al. [42] observed that increasing network width, thus capacity, leads to emergence of block-structure in non-robust networks. Thus we incrementally increase the width of robust Wide-ResNet architecture on CIFAR-10 dataset and measure the cross-layer representation similarity of benign features (Figure 3). Even after increasing network capacity by an order of magnitude, we do not see significant variation in the characteristics of robust network

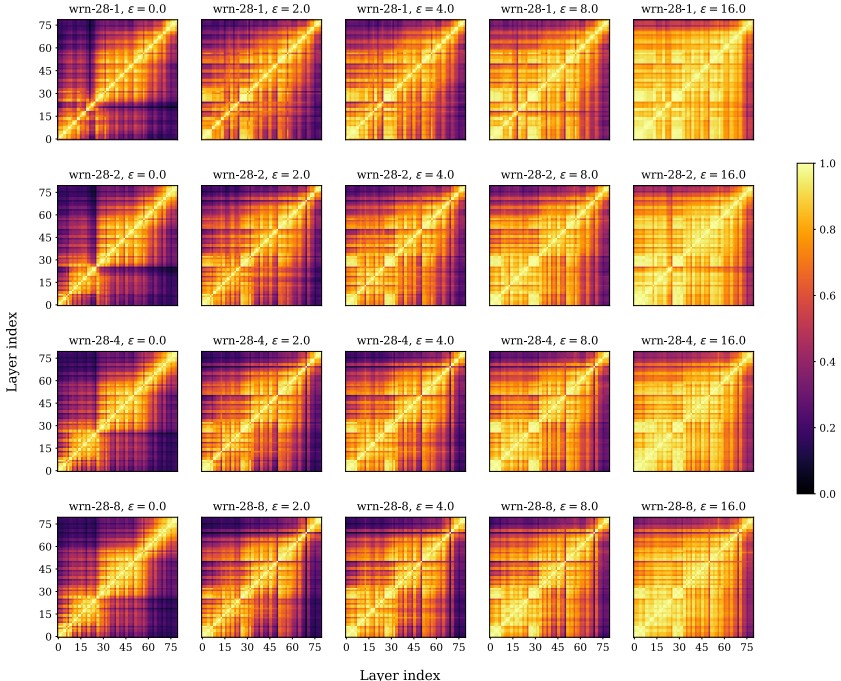

Figure 3: **Delving deeper into cross-layer similarity.** To better understand the unique cross-layer characteristics in benign representations of robust networks (Figure 2), we ablate along two main design choices: 1) Strength of adversarial attacks in robust training (by increasing perturbation budget ($\epsilon$)) and 2) Changing network size (by increasing network width). We observe that increasing the strength of adversarial perturbations in training predominantly leads to amplification of cross-layer similarity, i.e., higher degree of brightness in the plots. Increasing network capacity, even by an order of magnitude, does not substantially change this phenomenon. We used Wide-ResNet architectures with the CIFAR-10 dataset for this experiment.

representations. These results suggest that increasing width in robust networks doesn't lead to a drastic shift in similarity of internal layer representations.

## 4   Probing the impact of adversarial perturbations on representations

Robust training aims to learn both high quality benign and robust representations. In the previous section we characterized the impact of robust training on benign representations. In this section, we focus on understanding the characteristics of adversarial representations in robust networks. Detailed results, as well as ablations over architecture and threat model, are in Appendix C. We first analyze how adversarial perturbations distort the internal representations as we go deeper into the network, *i.e.*, compare adversarial and benign representations. Next, we analyze whether the distortions introduced by adversarial perturbations can also provide insights into why such perturbations transfer between different networks.

### 4.1   Impact of adversarial perturbations across layers in non-robust and robust networks

In this experiment, we measure the CKA similarity between benign image representations and corresponding adversarial image representations.

**Early layers are the least affected.** Adversarial perturbations are crafted to corrupt final layer representations. In our experiments, as expected, the cross-layer similarity score decays to near zero by the final layer (Figure 4). However, we observe another surprising trend in which adversarial representations share a high degree of similarity with a fraction (up to one-third for ImageNette and Imagewoof datasets) of early layers. We find this observation holds across architectures and budgets, indicating that earlier layers representations, even in benign networks, are robust to adversarial perturbations.

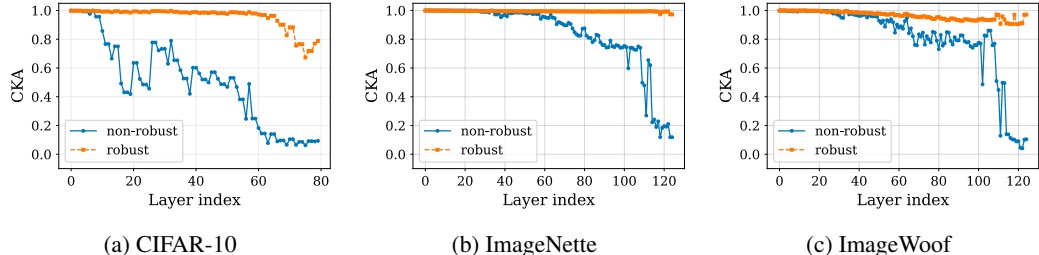

(a) CIFAR-10           (b) ImageNette           (c) ImageWoof

Figure 4: **Divergence of adversarial representations from benign ones.** We observe how adversarial perturbations, which are known to heavily impact model output, distort the internal representations in comparison to non-adversarial representations. we compare adversarial representations from corresponding layers in non-robust and robust networks. As expected, adversarial and benign representations in non-robust networks display high dissimilarity by the end of network. Unexpectedly, such divergence is very small for a non-trivial fraction of early layers (up to one-third). Robust training successfully reduces the impact of adversarial perturbations, thus leading to similar benign and adversarial representations in robust networks.

**Robust networks successfully prevent the distortion of internal representations.** For robustly trained networks, benign and robust representations maintain a high degree of similarity throughout the network, with divergence only occurring in the last 10 layers or so. In our ablation on network size (Figure 20 in Appendix C), we see that increasing the size of the network leads to increased differentiation, implying sufficient capacity for increased separability between benign and perturbed representations.

## 4.2 Understanding transferability through the lens of internal representation similarity

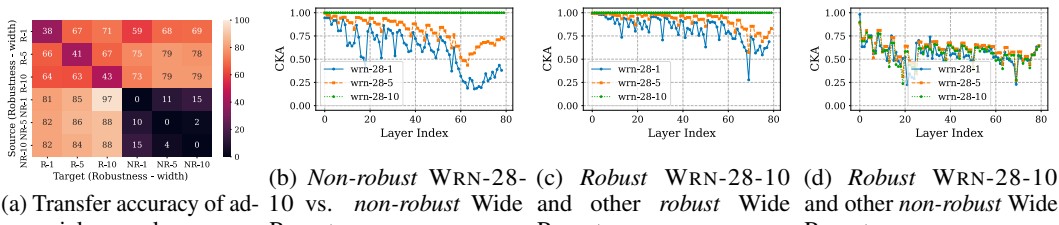

(a) Transfer accuracy of adversarial examples.    (b) *Non-robust* WRN-28-10 vs. *non-robust* Wide Resnets.    (c) *Robust* WRN-28-10 and other *robust* Wide Resnets.    (d) *Robust* WRN-28-10 and other *non-robust* Wide Resnets

Figure 5: **Analyzing cross-architecture representations to understand similarity.** *(a)* Measuring the robust accuracy of adversarial examples transferred between robust (R) and non-robust (NR) Wide-ResNet networks of varying width (e.g., R-5 refers to Robust WRN-28-5). Low robust accuracy implies high transferability. *(b, c, d)* We measure CKA between source and target networks, where these representations are extracted using adversarial examples generated for the source network. When both networks are trained in the same fashion (benign or robust), the layer wise similarity is generally high at earlier layers but drops off towards the end. In contrast, it drops right after the first layer from robust to non-robust network, a set of networks which also have least transferability.

It is well-known that adversarial examples are *transferable* from one architecture to another, i.e., adversarial examples generated on one network achieve the misclassification objective on others too. In Figure 5a, we measure the transferability between non-robust and robust networks. Intriguingly, as previously validated by [7], transferability is higher among some networks (non-robust to non-robust) while much poorer for others (robust to non-robust). High transferability indicates that the target network has learned a similar decision boundary to the source network. We aim to better understand this phenomenon by comparing the internal representations of source and target networks. We first generate adversarial examples, and their corresponding internal layer representations, from a source network (WRN-28-10 in this case). We compare them with internal layer representations of these

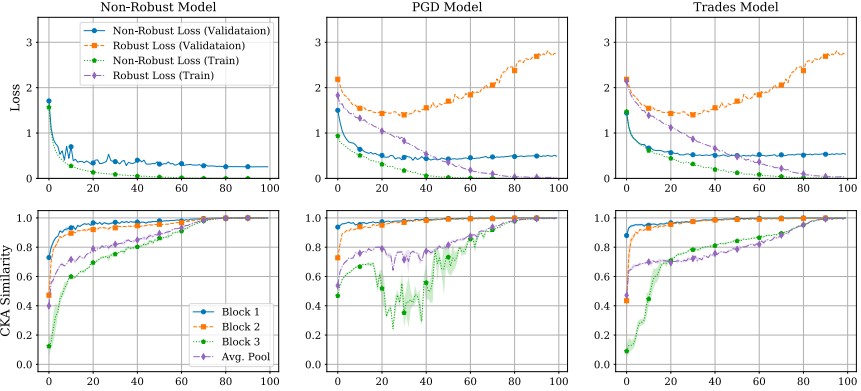

Figure 6: **Variation in convergence of different layers.** We compare representations from each epoch of training with the final learned representations of a network across both non-robust and robust (both PGD-based and TRADES) training techniques. We observe that representations obtained from the output of the first two blocks converge to their final state very early in the training process and have very little variation in both non-robust and robust network. We also find that later layers exhibit more complex behavior, with signs of overfitting when early stopping is not used, and a lack of stability from one epoch to the next. CKA values are averaged over three training runs with different random initializations.

adversarial examples on the target network. We consider transfer from non-robust to robust, robust to robust, and robust to non-robust networks [4] (Figure 5).

For transfer from non-robust to non-robust or robust-robust, where highest transferability occurs, we observe high degree of similarity between representations at early layers. It indicates that the transferred adversarial examples distort the internal representations in a similar fashion as source network. The most intriguing aspect in Figure 5a is extremely poor transferability from robust to non-robust networks. We establish that this is due to significant differences in representations where, even across identical architectures, we observe a large drop in CKA right after input layer. It shows that adversarial perturbations from robust nework distorts the internal representations of non-robust network in a strikingly different manner (even at very early layers), which likely renders them ineffective to cause an adversarial effect at the output of the network.

## 5 Evolution of internal representation dynamics during robust training

In this section, we track the evolution of representations learned by robust training methods over the course of training epochs in the optimization process. We study the following two convergence properties: 1) Rate of convergence at different layers, and 2) Stability of different layers throughout learning. As in earlier sections, we use CKA to measure similarity, but in this section we *compare the benign representations of the same layer across time*. We present our results for Wide-ResNet architecture in Figure 6, where we compare the representations at each epoch to the final epoch (epoch 100). We further investigate across architectures, layers, and attack budgets in Appendix D.

**Early layers converge faster than later layers.** Benign representations after the first two blocks of WRN-28-10 are highly similar to the representation obtained at the end of training, after just ten epochs (Figure 6). This is true for all the training methods considered. In Appendix D, we show this largely holds true across architectures and smaller budgets. Our observation relates to the intuition that early layers learn simpler features [33, 62, 64], thus are learned much faster than complex feature learned by later layers. Only with high budget (see $\epsilon = \frac{16}{255}$ in Fig. 26), i.e., stronger attacks, we observe a deviation from this trend where second block convergence is slower. We further validate this with layer-freezing experiments in Appendix D.5.

---

[4]Robust networks are known to be highly resilient to adversarial examples generated from non-robust networks (see Figure 5a).

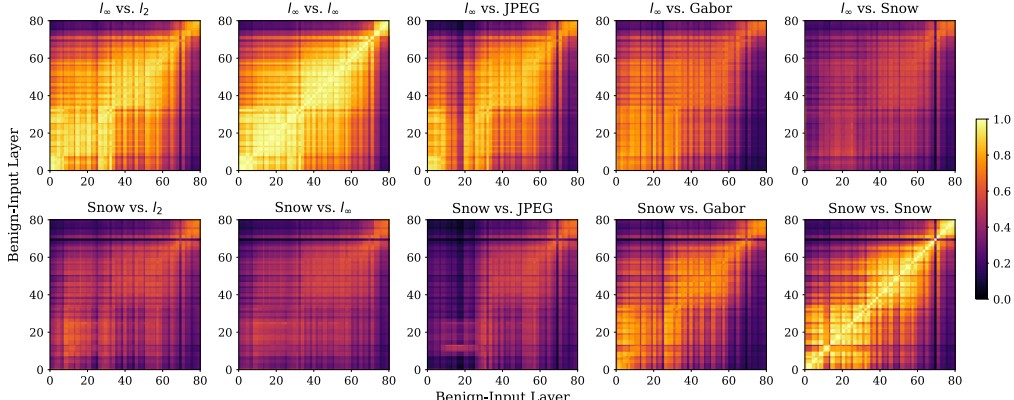

Figure 7: **Layer-wise similarity plots between different threat models**. The top row shows comparisons between all threat models and $l_\infty$, and the second row shows comparisons between all threat models and Snow. The $\ell_p$ threat models are similar to each other across layers. Among the other threat models, Snow and Gabor display the highest similarity, which is surprising due to large differences in their visual representations.

**Overfitting is predominantly visible in later layers.** From the cross-epoch heatmap of similarities between adversarial representations (Figure 25), we observe that later layer representations only exhibit similarity off-diagonal at the beginning (until epoch 30) and end of training. This clearly shows that multiple local minima are found during training. This is true to a less pronounced degree for benign representations. In Fig. 6, we also observe the adversarial validation loss increasing after around 30 epochs, while the training loss keeps decreasing, indicating overfitting. After this point in training, the CKA similarity between drops with respect to both the epoch 30 and final representation drops significantly in later layers, while the earlier layers remain unaffected. This strongly indicates that the later layers have overfit to the training data. While it is already known that adversarial loss induces overfitting at the output layer [48], our results show that this degree of overfitting is not present in all layers of the network. This finding supports previous work which shows that retraining just the final layer of a deep network can greatly increase its robustness to spurious correlations in training data [29]. We hypothesize that a layer-by-layer analysis of overfitting in networks may be useful in fine-tuning robust networks.

**Adversarial training reduces stability of internal representations during training.** While subsequent epoch representations tend to be very similar to each other in non-robust training, there there can be very large jumps in similarity between one epoch and the next, indicating a less stable convergence. This effect further accentuates when training with a higher perturbation budget or when training a larger network (Appendix D).

## 6   How does the threat model impact representations?

Throughout the paper, we have focused on a single threat model: $\ell_\infty$ perturbations. In this section, we extend our analysis of robust representations to understand what exactly the link is between the visual similarity of threat models and the representations of models trained to be robust to them. To this end, we consider *four* additional threat models: $\ell_2$, JPEG, Gabor and Snow [28] (visualized in Figure 8). We also examine the generalizability of our earlier results across these threat models. For space reasons, all associated figures from this section are in Appendix E.

**Do visually similar attacks lead to similar robust representations?** Representation similarity gives us a method by which we can compare how different classes of adversarial perturbations affect learned representations. Using CKA, we plotted the cross-layer similarity of robust WRN-28-10 networks adversarially trained against five different threat models ($\ell_2$, $\ell_\infty$, JPEG, Gabor, and Snow) (Figure 7). These comparisons lend credence to conventional knowledge about these threat models. The layerwise similarity plot between $l_2$ and $l_\infty$ robust models is strikingly similar to CKA plots between robust models trained on the same threat model. This mirrors theoretical results on correspondences

between $\ell_p$ threat models [55]. We also observe a higher average similarity when comparing an $\ell_\infty$ model with a JPEG model than with Snow or Gabor models. This makes intuitive sense, given the $\ell_\infty$ constraint inherent in the JPEG attack. When using CKA to compare against the Snow threat model, we observe that the highest average similarity is achieved with Gabor. This represents a novel insight into these threat classes, as correspondence between the two was not previously known, and may indicate the possibility of models jointly robust to these threat models.

**Alignment of budgets across different attacks.** CKA can also give insight into the varying effects of changing perturbation strengths within different threat models. Expanding on our experiments above (all figures are in Appendix E), we compared representations between different threat models while varying perturbation strengths. Between $\ell_\infty$ and Gabor models, the layerwise similarity structure is much more sensitive to changes in the strength used in $\ell_\infty$-robust training than in Gabor-robust training. For more closely related attacks, like $\ell_\infty$ and JPEG, we see similar degradations of the structure when changing the strength of either attack. Furthermore, the structure is largely preserved when the attack strengths are changed in tandem. These results may be useful for developing techniques for training models to be jointly-robust against different classes of attack.

**Layerwise structure across threat models.** We find that across all threat models except Snow, robust and benign activations exhibit high similarity (Figure 22). Changing budgets across the same threat model (Figure 30), we find that $\ell_p$ threat models' representations evolve more gradually.

# 7 Discussion

**Limitations.** The key limitations of our study arise from the fact that all discoveries are predicated on the effectiveness of the RS metric used to discern differences between representations. Further, the nature of our study can only unearth correlations between layer-wise and aggregate properties such as accuracy and loss. Establishing causal links needs further investigation, perhaps inspired by our findings as we discuss below. Finally, methods to improve robust training are beyond the scope of this paper. They need to be derived from a careful analysis of the observations in this paper for actionable changes to training strategies and architectures.

**Future Work.** Our observations highlight that robust networks could be improved in 3 key ways:

*1. Staggered freezing of layers during training:* Our results from § 5 indicate that early layers do not need to be updated post a few epochs of training, since their learned representations do not change much during training. Thus, knowledge of a network's training dynamics derived from CKA analysis can be used to increase the efficiency of training through the freezing of early layers.

*2. Increasing layer-wise differentiation during training*: Our results show there is a much greater degree of local similarity among learned representations for robust networks when compared to non-robust ones. This similarity also increases when the training budget is increased. We suspect this lack of layer-wise differentiation is part of the reason why robust networks do not achieve high accuracy on clean data. Using regularization methods that promote increased layer-wise differentiation during robust training may alleviate this issue, and is a compelling direction for future work.

*3. Choosing threat models for joint robust training*: Past work on training models jointly robust to multiple attacks has focused on different $\ell_p$ perturbations. We posit that our analysis of the similarity of representations obtained from different threat models can be utilized to determine sets of threat models for which joint robustness is possible, and if the model has sufficient capacity. Further, the layer-wise analysis can be used to add appropriate regularizers to ensure convergence, an issue exacerbated by the presence of multiple types of adversarial examples.

## Acknowledgments and Disclosure of Funding

We thank our anonymous reviewers for their insightful feedback. CC, ANB, BYZ and HZ were supported in part by NSF grants CNS-1949650, CNS-1923778, CNS-1705042, the C3.ai DTI, and the DARPA GARD program. VS and PM were supported in part by NSF grants CNS-1553437 and CNS-1704105, the ARL's A2I2, the ONR Young Investigator Award, the ARO Young Investigator Prize, Schmidt DataX award, Princeton E-ffiliates Award, and Princeton Gordon Y. S. Wu Fellowship. Any opinions, findings, and conclusions or recommendations expressed in this material are those of the authors and do not necessarily reflect the views of any funding agencies.

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
