In this Appendix, we aim to further validate the discoveries from the main body of the paper by conducting additional experiments. Our main focus is to ablate across architectures, representation similarity metrics and threat models to understand the generalizability of the results presented in the main body. For specific experiments, we also vary other parameters such as datasets as necessary.

The Appendix is organized as follows:

1. Further background and related work (Section A)
2. Exploring benign representations from robust networks (Section B)
3. Analysis of adversarially perturbed representations (Section C)
4. Impact of training parameters on layer-wise convergence (Section D)
5. Alignment of different threat models (Section E)

## A   Additional background and setup details

### A.1   Representation similarity metrics

**Canonical Correlation Analysis (CCA):** Given activation matrices $X \in \mathbb{R}^{n \times p_1}, Y \in \mathbb{R}^{n \times p_2}$, canonical correlation analysis involves finding the orthogonal bases $w_X^i, w_Y^i$ such that, after the matrices are projected onto their respective bases, correlation between the matrices is maximized [31]. The relevant summary statistic for CCA is $\bar{\rho}_{CCA}$, which is defined to be the mean of the canonical correlation coefficients $\rho_i$, for $1 \leq i \leq p_1$. $\rho_i$ can be computed by

$$
\begin{aligned}
\rho_i = \max_{w_X^i, w_Y^i} \ &\mathrm{corr}(X w_X^i, Y w_Y^i) \\
s.t. \quad &\forall_{j<i} X w_X^i \perp X w_X^j \\
&\forall_{j<i} Y w_Y^i \perp Y w_Y^j
\end{aligned}
\tag{4}
$$

In order to improve the robustness of CCA, **Singular Vector Canonical Correlation Analysis (SVCCA)** has been proposed by Raghu et al. [44]. Rather than compute CCA on $X$ and $Y$, SVCCA first computes the singular vector decomposition of $X$ and $Y$ to find representations $X'$ and $Y'$ whose principle components explain a fixed proportion of the variance of $X$ and $Y$. Standard CCA is then performed on $X'$ and $Y'$.

**Orthogonal Procrustes:** The solution to the Orthogonal Procrustes problem is the left rotation of $X$ that minimizes the Frobenius norm between the rotated $X$ and $Y$. Formally, the problem is defined as

$$
\min_R \|Y - RX\|_F^2, \ s.t. \ R^\mathsf{T} R = I
\tag{5}
$$

The Orthogonal Procrustes distance between $X$ and $Y$ is defined as

$$
d_{\mathrm{Proc}}(X, Y) = \|X\|_F^2 + \|Y\|_F^2 - 2\|X^\mathsf{T} Y\|_*,
\tag{6}
$$

where $\|\cdot\|_*$ is the nuclear norm [14]. When $X$ and $Y$ are normalized, this distance metric lies in the range $[0, 2]$. For our purposes, we compute the Orthogonal Procrustes similarity between normalized matrices $X$ and $Y$, defined to be

$$
S_{\mathrm{Proc}}(X, Y) = 2 - d_{\mathrm{Proc}}(X, Y)
\tag{7}
$$

### A.2   Threat models

Most of our robust models were trained and evaluated on $\ell_\infty$ adversarial images. To supplement our results, we also ran experiments using conventional $\ell_2$ attacks, as well as three adversarial attacks introduced by Kang et al. [28]: JPEG, Snow, and Gabor. The JPEG attack adds $\ell_p$ bounded adversarial noise to the JPEG-encoded space of images, rather than the pixel space of images. The Snow attack adds adversarially optimized visual occlusions to images that simulate snowfall. The Gabor attack optimizes the parameters of a Gabor kernel that is used to noise the image. Visual examples of these attacks are shown in Figure 8. Unless otherwise noted, for our CIFAR-10 experiments we run the $\ell_2$ attack with $\epsilon = 1$, the JPEG attack with $\epsilon = 25$, the Snow attack with $\epsilon = 2$, and the Gabor attack with $\epsilon = 75$.

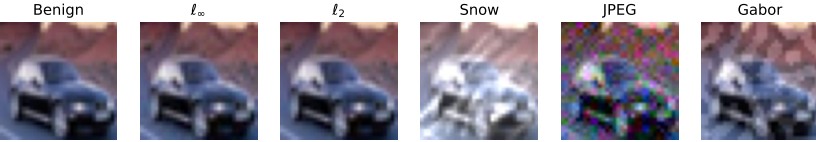

| Benign | $\ell_\infty$ | $\ell_2$ | Snow | JPEG | Gabor |

Figure 8: Adversarial examples generated by the threat models used in this paper.

| | Benign Images | | Adversarial Images | |
|---|---|---|---|---|
| Threat Model | Train Accuracy | Val. Accuracy | Train Accuracy | Val. Accuracy |
| Base | 100.00 | 94.92 | - | - |
| $\ell_\infty$ ($\epsilon = \frac{8}{255}$) | 100.00 | 86.36 | 99.03 | 46.34 |
| $\ell_2$ ($\epsilon = 1$) | 100.00 | 86.49 | 99.40 | 45.76 |
| JPEG ($\epsilon = 25$) | 100.00 | 84.65 | 91.98 | 45.52 |
| Gabor ($\epsilon = 75$) | 100.00 | 93.12 | 99.83 | 92.47 |
| Snow ($\epsilon = 2$) | 100.00 | 90.23 | 95.72 | 67.35 |

Table 2: Training and validation accuracies of the WRN-28-10 models used in this paper's experiments. The threat model represents the attack that each model was trained to be robust against, and adversarial images were generated using the same threat model and attack strength as was used in training.

## A.3 Related Work

**Tools to probe neural networks.** These can be sorted broadly into two categories: i) *sample-based* and ii) *distribution-based* tools. Sample-based tools include techniques like input saliency maps [52], Grad-CAM [50], integrated gradients [53] etc. which are focused on interpreting the output of neural networks for specific inputs. On the other hand, loss surface visualizations [62, 41, 61] and representation similarity metrics [16, 31, 2, 44, 9] consider aggregate network properties derived from a set of samples.

**Visualizing and validating network properties:** Gilboa and Gur-Ari [19] use activation atlases to argue that the learned representations of wider networks are 'more informative' than shallower ones. To validate this assertion, they fine-tune a linear layer for a new task and find the wider representations perform better. Raghu et al. [45] study how transformers and CNNs learn features differently, Grigg et al. [21] analyze the differences between supervised and self-supservised representations and Kornblith et al. [30] consider the link between loss functions and feature transfer. Gavrikov and

| | Non-Robust Model | | $\ell_\infty$-Robust Model | | | |
|---|---|---|---|---|---|---|
| | Benign Images | | Benign Images | | Adversarial Images | |
| Width | Train Acc. | Val. Acc. | Train Acc. | Val. Acc. | Train Acc. | Val. Acc. |
| 1 | 99.68 | 91.23 | 89.21 | 82.39 | 50.90 | 42.92 |
| 2 | 99.97 | 93.44 | 97.73 | 84.91 | 66.91 | 44.02 |
| 3 | 99.99 | 94.18 | 99.72 | 85.53 | 80.78 | 43.74 |
| 4 | 100.00 | 94.33 | 99.98 | 85.46 | 89.64 | 44.13 |
| 5 | 100.00 | 94.66 | 100.00 | 85.97 | 93.42 | 45.38 |
| 6 | 100.00 | 94.86 | 100.00 | 86.40 | 95.79 | 45.70 |
| 7 | 100.00 | 94.59 | 100.00 | 86.87 | 97.38 | 46.79 |
| 8 | 100.00 | 94.50 | 100.00 | 86.90 | 98.24 | 47.15 |
| 9 | 100.00 | 94.98 | 100.00 | 87.39 | 98.68 | 48.11 |
| 10 | 100.00 | 94.92 | 100.00 | 87.59 | 98.90 | 48.89 |

Table 3: Training and validation accuracies of the WRN-28-N models used in this paper's experiments, where N is the width factor. Robust models were trained on the $\ell_\infty$ threat model with $\epsilon = \frac{8}{255}$.

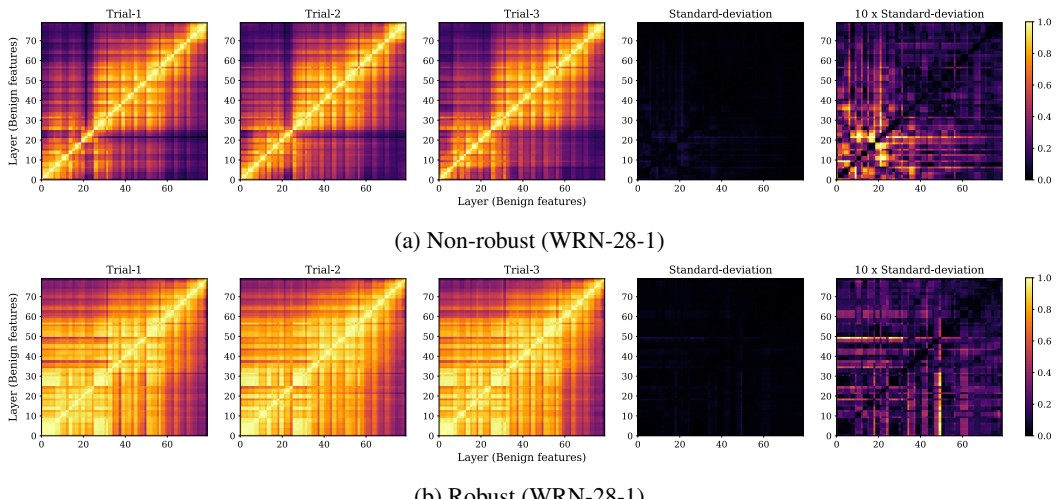

(a) Non-robust (WRN-28-1)

(b) Robust (WRN-28-1)

Figure 9: **Standard deviation of CKA across different random initializations.** We evaluate the CKA of three random initializations of non-robust and robust WRN-28-1 models, independently, and report the standard deviation. As shown above, the standard deviation is small, verifying that CKA is largely invariant to the random initialization of a network.

Keuper [18] use convolutional filters to understand the impact of robust training while Ilyas et al. [27] visualize what input features are needed for robust training.

## A.4 Stability of CKA over different random initializations

Prior work has shown that CKA results are consistent across different random initializations [31]. We validate this by reporting the standard deviation of a cross-layer CKA computation in Figures 9a and 9b.

## A.5 Stability of Online CKA over identically-trained networks

In Figures 10a and 10b, we show that the standard deviation of the unbiased estimate of CKA provided by our online CKA algorithm is very small.

# B How are benign representations from robust networks different?

In this section, we provide additional details on how the benign representations from robustly trained networks differ from those obtained from non-robust networks (see §3 of the main paper). We ablate across RS metrics (appendix B.1), datasets (appendix B.2), robust training methods (appendix B.3) and architectures (appendix B.4). We also delve deeper into the link between RS metrics and classification accuracy in appendix B.7.

## B.1 Using other representation similarity metrics

In Figure 11, we plot the layerwise similarity between the layers for benign representations from both a non-robust and robust model. Due to computational constraints when computing SVCCA and Orthogonal Procrustes, comparisons are shown using ResNet18 models, instead of the Wide ResNet models used throughout much of the rest of the paper. We note that the trend we observe of increased cross-layer similarity for robustly trained networks appears regardless of the RS metric used. Further, CKA leads to the most visually distinct structure. The broad takeaways from the different RS metrics are aligned enough that we focus on CKA for the remainder of the results in this Appendix.

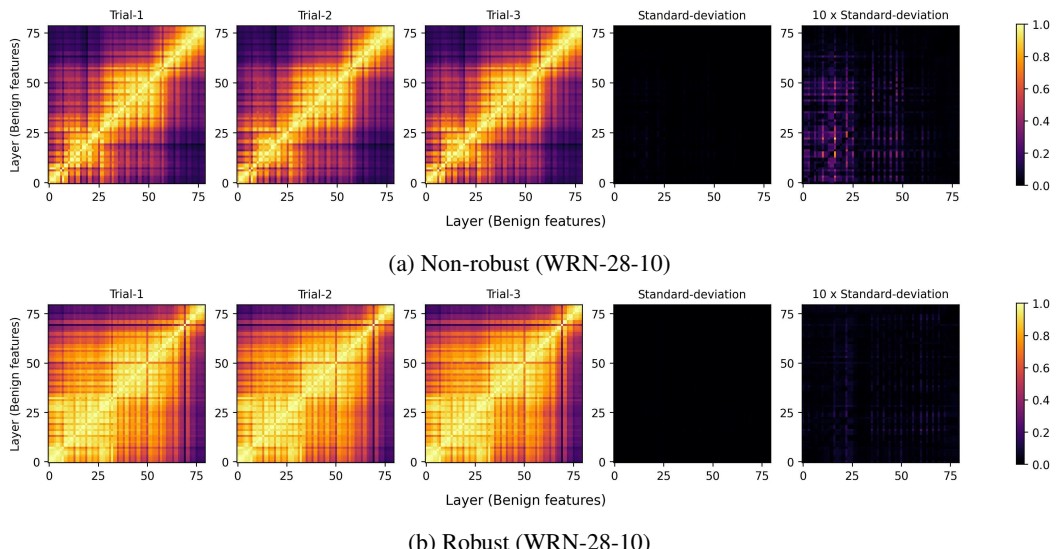

(a) Non-robust (WRN-28-10)

(b) Robust (WRN-28-10)

Figure 10: **Standard deviation of online CKA.** We evaluate the *online* CKA of a non-robust and robust WRN-28-10 model three times, independently, and report the standard deviation. As shown above, the standard deviation is very small, indicating that the insights from CKA are stable across individual evaluations.

## B.2  Using other datasets

In Figures 12 and 13, we plot the layerwise similarities using CKA for ResNets trained on the ImageNette and Imagewoof datasets, both of which are subsets of the Imagenet dataset. These expand upon the results from Figure 2 of the main body, and show the stark differences between the representations obtained from non-robust and robust networks. The cross-layer similarities are far more pronounced for robust networks across datasets and architectures.

## B.3  Comparing across robust training methods

In Fig. 14, we compare the benign representations from models robustly trained using two different robust training techniques: PGD-based adversarial training and TRADES. Both methods lead to very similar robust representations, and exhibit the same long-range similarity across layers.

In Figure 15, we compute cross-layer CKA similarities of seven robust networks from the Robust-Bench leaderboard. We find that, across training algorithms and architecture, these networks all display the lack of block structure that we saw in our own PGD-trained robust models.

## B.4  Comparing across architectures

Recent work [26] has studied the influence of depth and width on adversarial robustness. We extend this investigation by comparing the benign representations obtained from Wide-ResNet models trained with different widths in Figure 16 with both non-robust and robust training. A few key observations stand out: i) early layers in different blocks are very similar across different widths for robust training, as compared to benign training, ii) increasing width only leads to large variations in learned representations in the last block for robust training, and iii) benign networks of different widths are similar deeper in the network, and different early on, with the opposite trend for robust networks. These results, taken together suggest that increased width early on in the network may not be particularly useful, but deeper in the network, leads to divergent representations, which can be investigated further to synthesize robust architectures.

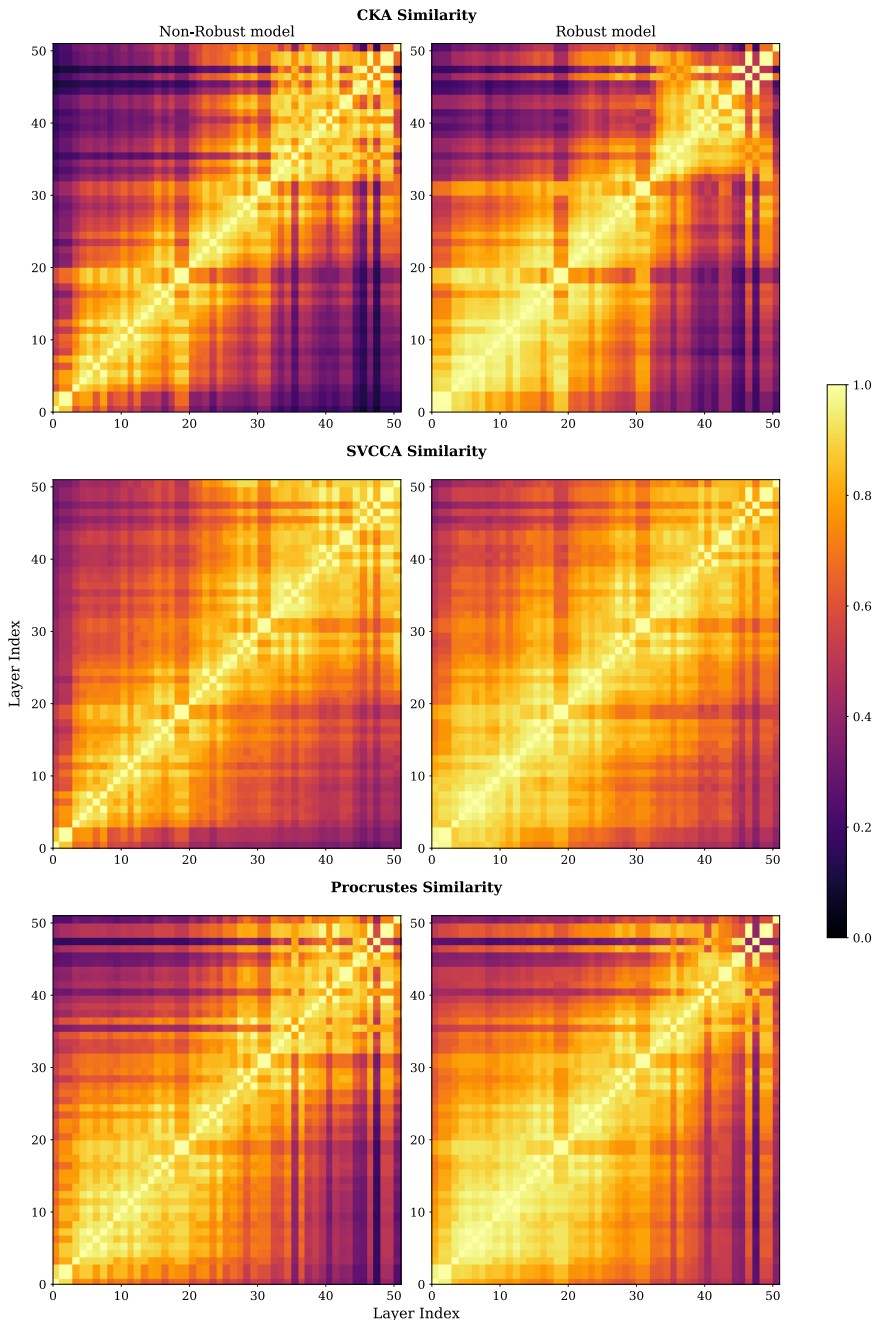

Figure 11: **Layerwise similarity plots of non-robust and robust ResNet18 models**. 3 different RS metrics are used: CKA, SVCCA, and Orthogonal Procrustes similarity over benign representations.

## B.5 Comparing across attacks

In Figure 17, we compute cross-layer CKA similarity for benign and robust models using three different adversarial attack algorithms (Fast Gradient Sign Method, Projected Gradient Descent, and Auto-Projected Gradient Descent). Our results indicate that choice of attack causes little change in cross-layer similarity, with any differences likely being due to the relative strength of each attack.

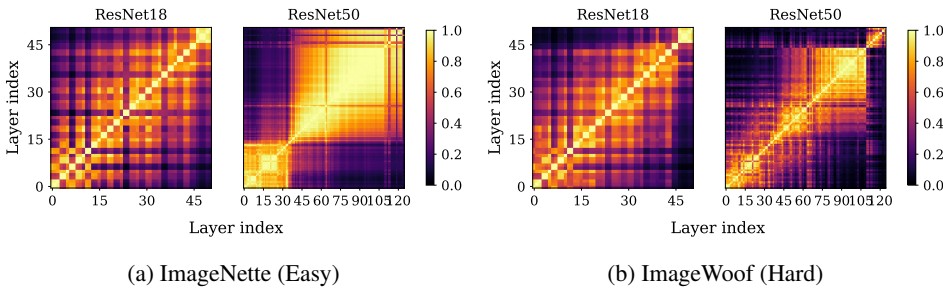

(a) ImageNette (Easy)                    (b) ImageWoof (Hard)

Figure 12: Cross layer CKA for *non-robust* network (using benign features)

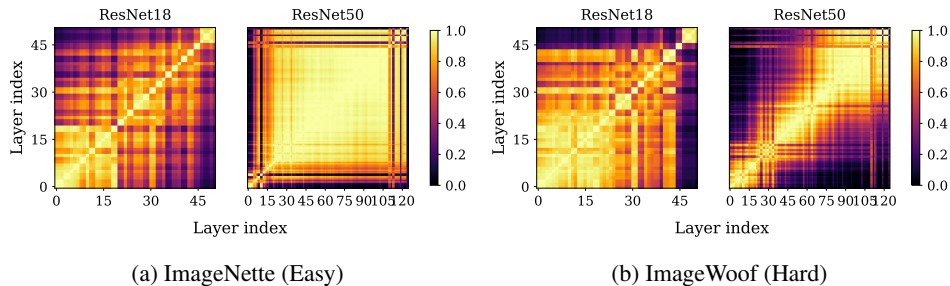

(a) ImageNette (Easy)                    (b) ImageWoof (Hard)

Figure 13: Cross layer CKA for *robust* network (using benign features)

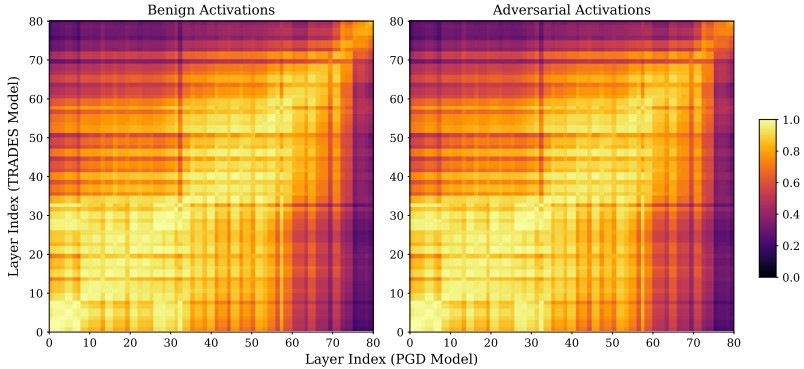

Figure 14: **PGD-based adversarial training and TRADES produce similar final representations.** Layerwise similarity between benign activations of a TRADES-trained WRN-28-5 model and a PGD-trained WRN-28-5 model.

### B.6   Common Image Corruptions

Beyond adversarial perturbations, our methodology can also extend to other types of image perturbations. Results from networks trained to be robust to common types of non-adversarial corruptions are included in Figure 18. We find that these corruptions have a much less pronounced effect on learned representations than adversarial perturbations.

### B.7   Delving deeper into CKA

**Class-aware representation similarity.** Standard RS metrics don't consider data labels, thus similarity scores doesn't provide insights in how individual class data impacts it. As an example, we consider Linear CKA (Equation 3 from the main body), where the similarity is dependent on the dot-product of vectorized Gram matrices ($\boldsymbol{K}, \boldsymbol{L}$). Two factors contribute to the similarity of gram matrices: 1) Intra-class similarity ($C_1$): $\sum_{i,j,y_i=y_j} K_{ij}L_{ij}$, 2) Inter-class similarity ($C_2$): $\sum_{i,j,y_i \neq y_j} K_{ij}L_{ij}$. When measured between identical features, linear CKA is 1, i.e., $C_1/(C_1 +$

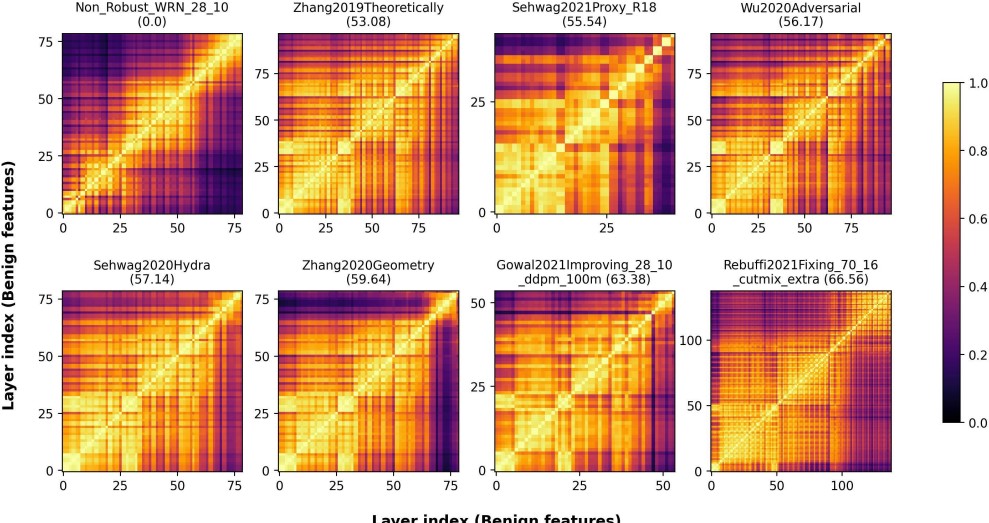

Figure 15: **Impact of adversarial training techniques (training algorithms, network architecture, and extra data) on our findings.**. Adversarial training performance has significantly improved over the last couple of years, where factors like better adversarial training techniques, better architectures, and extra training data helped. Our goal is to investigate whether our findings remain agnostic to these choices. We consider seven robust networks from the RobustBench leaderboard, which have varying performance and training setups. In each, we measure the cross-layer similarity of benign image features and find that the most robust networks do exhibit strikingly different cross-layer similarities than a non-robust network. Even when the network capacity is increased by an order of magnitude (e.g, in WRN-70-16), robust networks don't show as striking a block structure as a non-robust network. Each figure title corresponds to its key in the RobustBench leaderboard and the value in the title corresponds to the network robust accuracy on the CIFAR-10 dataset.

$C_2) + C_2/(C_1 + C_2)$. Under a near uniform distribution of class labels, only a small fraction of entries in gram matrices contribute to intra-class similarity ($C_1$), as there are a lot more cross terms in the CKA computation. However, when experimentally measuring the influence of both terms, i.e., we find that the contribution of intra-class CKA is very high, and it increases as we go deeper into the network. This is likely because separability of feature improves deeper in the network with similar class features being clustered together, which naturally increases the magnitude of $C_1$ in the similarity score.

We hypothesized that robust networks would have lower intra-class similarity as compared to non-robust networks, which would explain their lower classification accuracy on benign data. While this holds for the CIFAR-10 and ImageNette, for Imagewoof, robust networks have a much higher intra-class contribution. We leave a detailed investigation for future work.

## C   Adversarially Perturbed Representations

In this section, we add further results to back up the insights about the impact of adversarially perturbed data on resulting representations from §4.

### C.1   Comparing benign and perturbed representations

In Figure 20, we can see, across, architectures, benign and perturbed representations differ at all layers except the first few for non-robust networks. In addition, for robustly trained networks, increasing width increases the divergence between benign and perturbed representations. This may indicate the presence of increased capacity to learn different representations for benign and adversarial inputs.

Across different threat models (Figure 22), we find robustly trained networks learn aligned benign and perturbed representations. This indicates that, in general, for most threat models, robust training

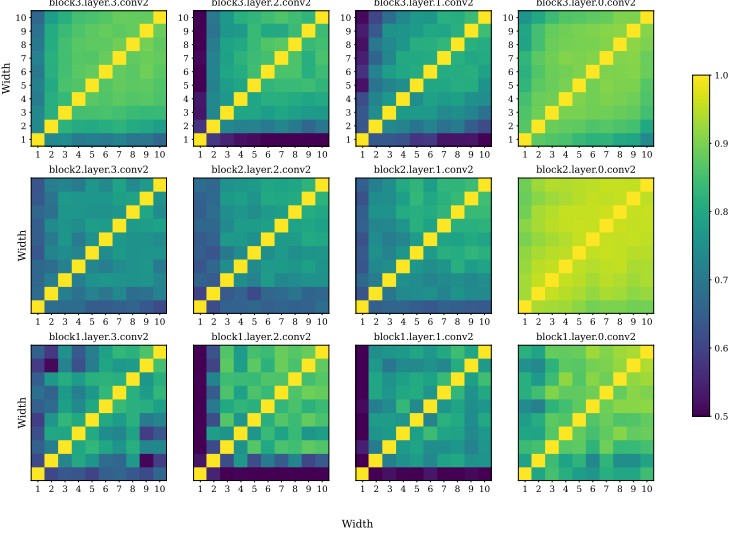

(a) *Non-robust* networks. Features extracted over *benign* images.

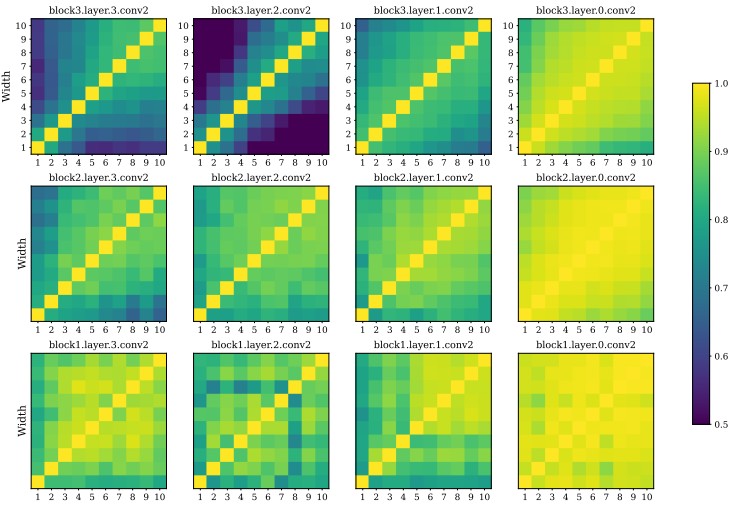

(b) *Robust* networks. Features extracted over *benign* images.

Figure 16: **Cross-model CKA between Wide ResNet models of different widths**. Data is presented across 12 convolutional layers, starting from the last convolutional layer, i.e., block3.layer3.conv2 and moving backward in the network. Robust networks have a high degree of similarity across widths in earlier layers but large variation in deeper layers, which is the opposite trend to that observed for non-robust networks.

is effective at ensuring that perturbed representations do not differ too significantly from benign ones, unlike for non-robust networks.

## C.2 Adversarially perturbed representations of non-robust networks

When analyzing the architectural impacts on learned representations using benign data, Nguyen et al. [42] note the emergence of a block structure with increased width and depth. We find that a much stronger block structure emerges even for low-width networks when using adversarial inputs (Fig. 23). This implies that perturbations force representations within a residual block to have higher similarity than outside the block. The block structure has been linked to the capacity of models, with Nguyen et al. [42] noting that the 'block structure arises in models that are heavily overparametrized relative to the training dataset'. In this light, adversarial examples have a peculiar

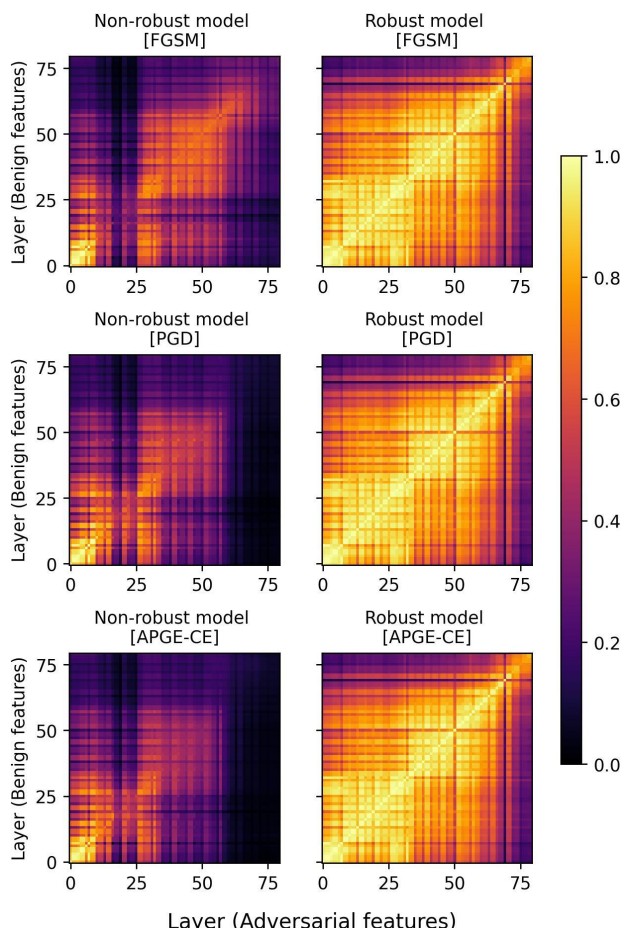

Figure 17: **Choice of adversarial attack**. We consider three different attacks, namely FGSM, PGD, and APGD-CE (arranged in order of attack strength) in this experiment. With each attack, we measure the cross-layer similarity between adversarial and benign representations. Our results show that such cross-layer similarity shows a similar trend across attacks, showing that our observation is agnostic to attack design choice.

impact on the representations of benign models, since the block structure is more clearly visible for adversarial inputs, but for models which do not classify the inputs correctly. This indicates that local similarity is being exhibited and is a different phenomenon compared to block structure emergence for well-trained models.

Figure 24 shows the stark difference between perturbed features for robust and non-robust networks. This is similar to the lack of similarity for benign inputs to these networks.

# D    Evolution over Training

In this section, we study the similarity between the activations produced by models after each epoch of training. We present our results in two formats, time series and heatmaps. For ease of understanding, the time series corresponds to a single row of the heatmap, where the representation being compared to is held constant.

## D.1    Impact of Model Capacity

In Figure 25, we plot the pairwise similarity between the representations of a single layer after each epoch of training. Natural training behaves as we would expect it to: a layer's representations at a

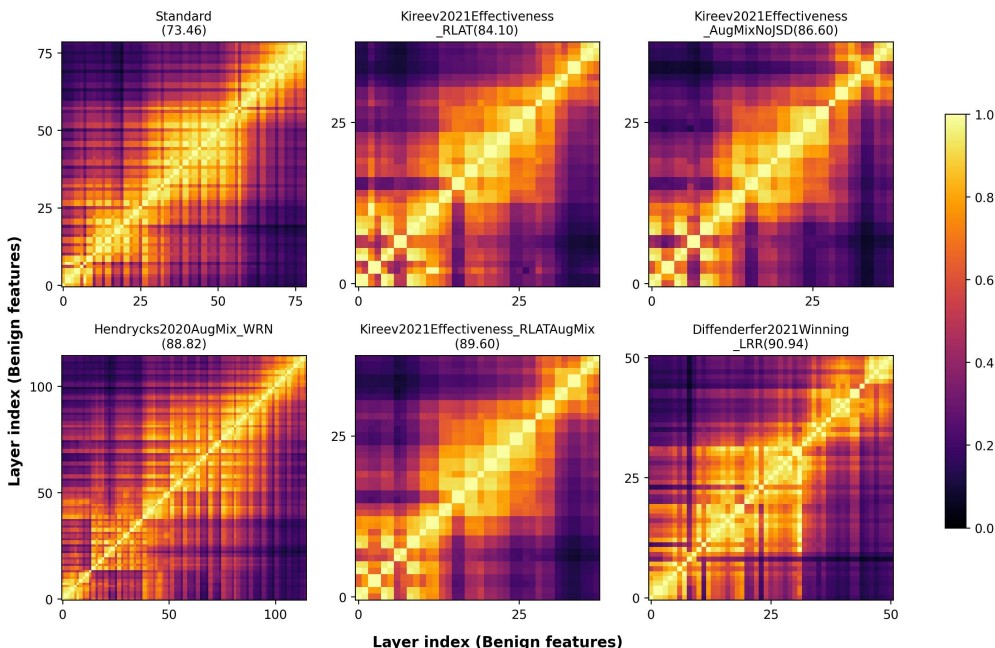

Figure 18: **Robustness to image corruptions**. While the main focus of this work is to investigate robust representation for worst-case adversarial attacks, we also investigate representations learned by networks robust to only common corruptions (such as photometric and weather changes). We consider multiple networks from the RobustBench [7] leaderboard that achieve better performance than a non-robust network on the CIFAR-C [23] dataset. We find that the difference between representations of the robust and non-robust networks is not as pronounced in the case of common corruption robustness. It suggests that the divergence of representations in non-robust and robust networks further depends on the strength of attack threat model.

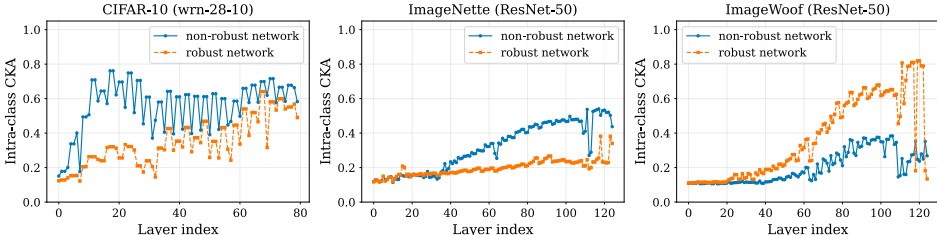

Figure 19: **Intra-class similarity.** We find that the contribution of intra-class similarity, measured by intra-class CKA, increases to non-trivial fraction, as we go deeper into the network.

given epoch are most similar to those from nearby epochs, and similarity degrades as the distance between epochs increases. Higher average similarity between later epochs implies convergence to a final learned representation. This pattern is also observed in the robust training of low-capacity networks, but as capacity increases starkly different dynamics emerge. In high-capacity robust networks, we observe similarity start to decrease after about 30 epochs of training, only to increase again later on. This pattern is observed for both benign and adversarial activations.

## D.2 Adversarial Training and Overfitting

In Figure 26, we plot the similarity to the final learned representations of the representations extracted from each of the three Wide ResNet blocks (block1, block2, block3) and the final average pooling layer before classification (avgpool) after each epoch of training. We also plot the validation loss of each model for comparison. We find that for models trained at $\epsilon = \frac{8}{255}$, instability in the later layers is not observed until the model begins to overfit on adversarial examples. At high perturbation

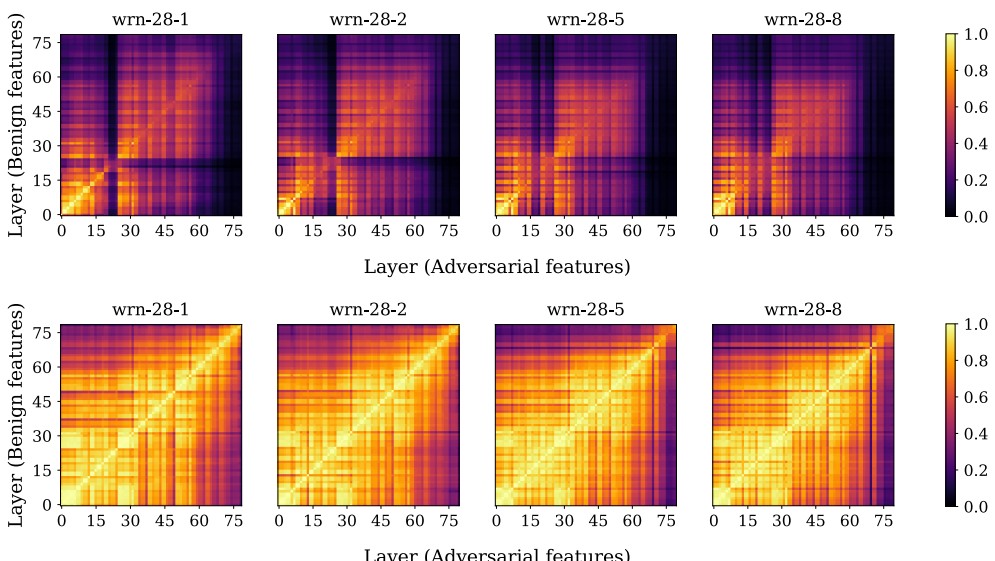

Figure 20: **Adversarial examples cause representation drift in later layers.** In the similarity plots of non-robust network representations (*top row*), adversarial and benign features display a similar structure to plots comparing just benign representations of non-robust models only in the early layers. In later layers, adversarial representations are very dissimilar from the benign representation of all layers. The similarity plots of robust networks (*bottom row*) are quite similar to plots comparing just benign representations of robust models, suggesting that robust networks learn similar representations for benign and adversarial data.

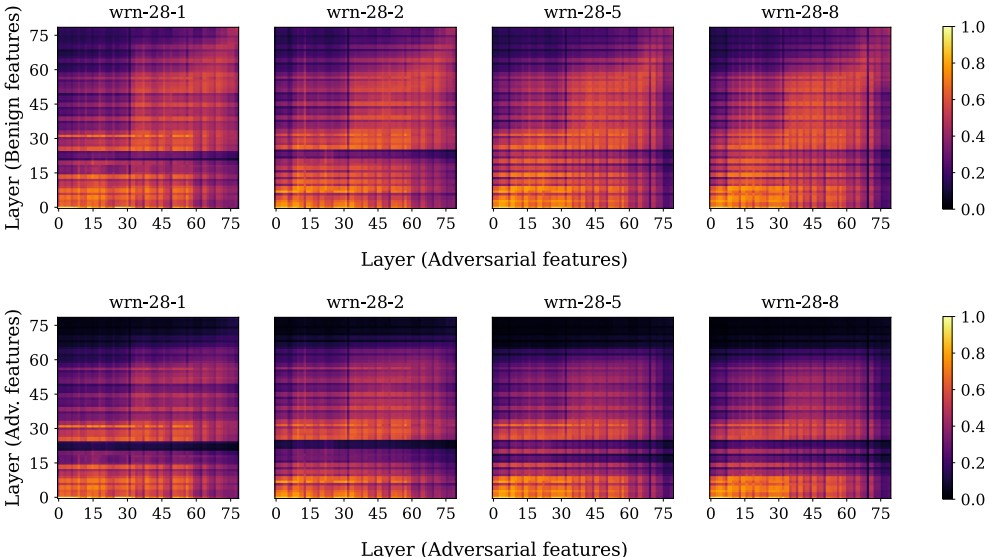

Figure 21: **Non-robust and robust models differ substantially across layers for both benign and perturbed inputs.** Cross layer CKA between a non-robust model (Y-axis) and robust model (X-axis) using perturbed and non-perturbed features.

strengths, training instability is observed at earlier layers and earlier in training. Training of the lowest capacity model is stable at all perturbation strengths, a trend also observed in Figure 25.

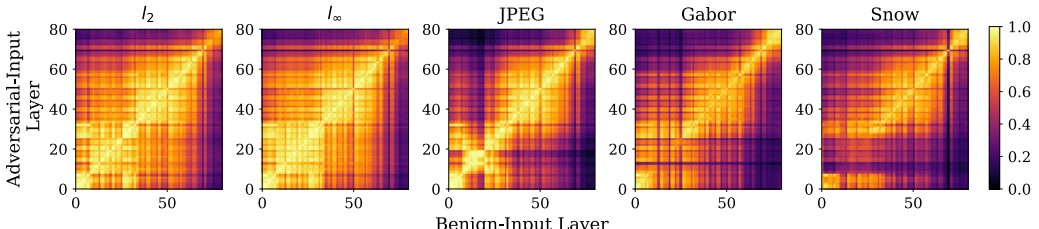

Figure 22: **Robustly trained models across threat models have aligned benign and perturbed representations.** Except the 'Snow' threat model, all the others have identical benign and perturbed representations across layers.

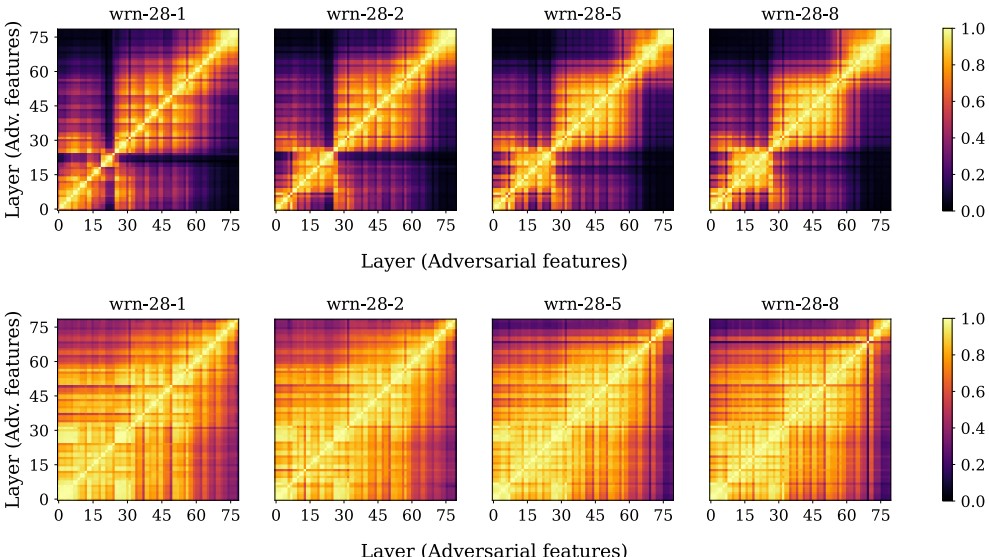

Figure 23: **Adversarial examples have different impacts on the representations of benign and robust models.** In benign models (top row), the layerwise similarity between adversarial activations displays a similar structure to that of benign activations, except the layers with dissimilar benign activations appear to have even more dissimilar robust activations. This implies significant changes in adversarial activations are taking place in between the self similar layer "blocks". This pattern is largely absent from robust models (bottom row), which display similar patterns for both benign and adversarial inputs.

### D.3 Different Training Methods

We also sought to test whether more principled robust training methods, such as those proposed by Zhang et al. [65], would affect the patterns in similarity we observed in this section. In Figure 6, we compare benign training, PGD training, and TRADES training in a similar manner to Figure 26. The patterns we observed with PGD training disappeared when training with TRADES, highlighting the sensitivity of CKA to meaningful changes in training dynamics.

### D.4 Training Dynamics of Different Threat Models

We extend our analyses from Section 5 to look at the Gabor and JPEG threat models. Results are plotted in Figure 27. We find that there are noticeable differences, especially in regards to overfitting observed in in $\ell_\infty$ training. However, across all threat models, we see that early layers are less affected by adversarial training.

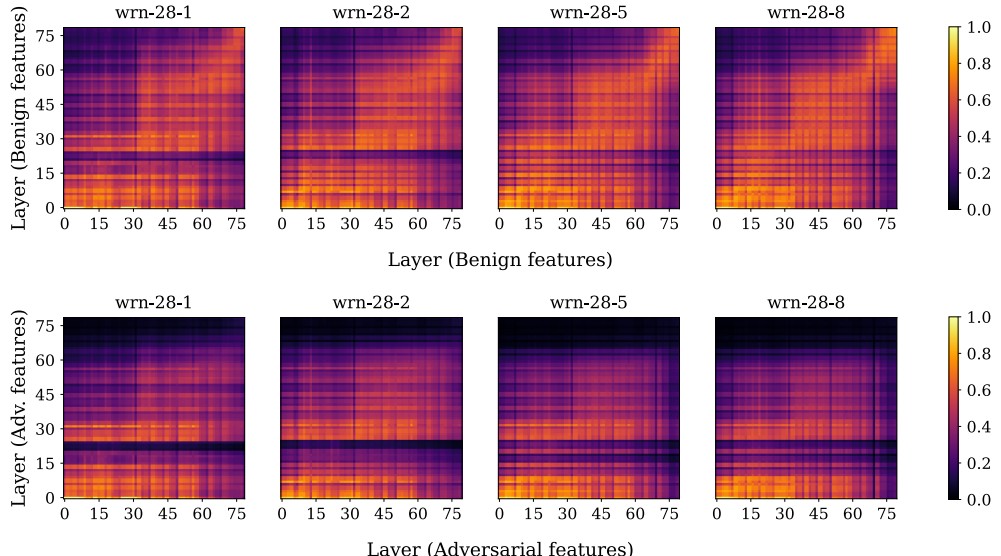

Figure 24: **Perturbed representations for robust and non-robust networks differ across all layers.**
Cross layer CKA between a non-robust model (y-axis) and robust model (x-axis) using perturbed
features.

### D.5 Layer Freezing

Based off of the results of our experiments looking into the CKA analysis of training dynamics, we
test whether those findings can be used to increase the efficiency of training. To test this, we froze
the first block of a WRN-28-5 at different points in the first 40 epochs of training and recorded the
maximum adversarial validation accuracy achieved over 100 epochs of training. As shown in Figure
29, when increasing the epoch freezing occurs at, accuracy starts off low (below 43%) and steadily
increases until epoch 20, after which it levels off around 46-47%. This matches the trend of the CKA
similarity convergence of convolutional layers within block 1 of a WRN-28-5 network, as shown in
Figure 28. These results suggest that with knowledge of a network's training dynamics derived from
CKA analysis, we can take a principled approach to layer freezing while maintaining a relatively
high degree of accuracy.

## E Threat models

In this section, we further investigate the impact of threat model on learned representations (see §7 in
the main body).

### E.1 Impact of budget within a threat model

In Figure 30, we show that even within a single attack, changing the budget leads to drastic changes
in the representations at deeper layers.

### E.2 Cross-threat model similarity

In Figure 31, we present a pairwise comparison of benign representations extracted from robust
models trained on each of our threat models. This expands upon Figure 7 from the main body and
demonstrates that while visual similarity leads to similar robust representations, the converse is not
necessarily true. Snow and Gabor have similar representations in spite of being extremely visually
different threat models (Fig. 8), demonstrating the value of using RS metrics to evaluate the viability
of joint robust training.

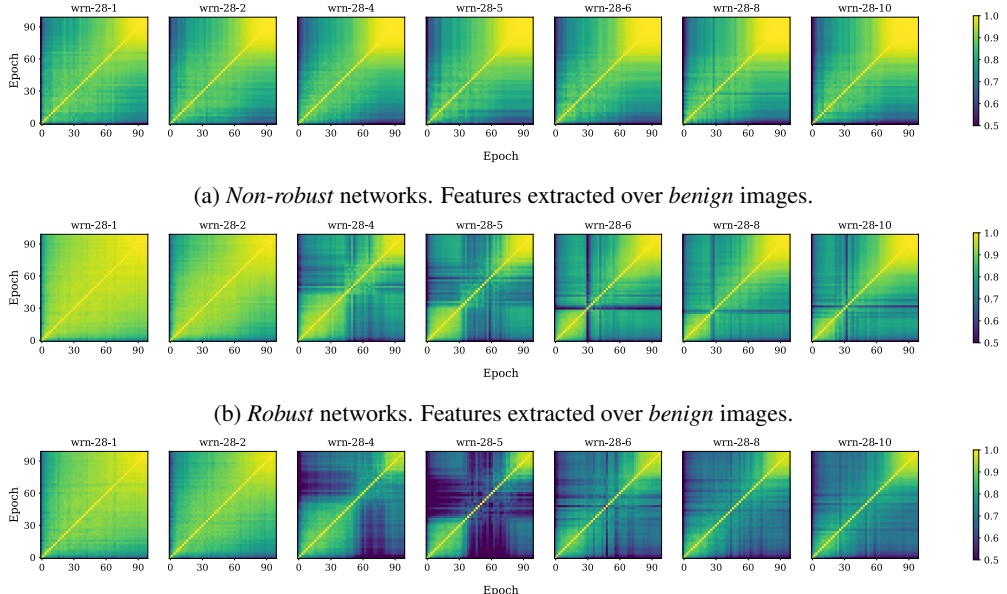

(a) *Non-robust* networks. Features extracted over *benign* images.

(b) *Robust* networks. Features extracted over *benign* images.

(c) *Robust* networks. Features extracted over *adversarial* images.

Figure 25: **Stability of robust training is impacted greatly by model capacity.** Each plot shows the CKA similarity between the activations of the final layer of a Wide ResNet feature extractor at each epoch of its 100-epoch training. While the stability of non-robust training is largely unaffected by capacity, the robust training of models above a certain capacity causes representations to be learned in the middle of training that are starkly different from the initial and final representations. This effect is even more pronounced when comparing adversarial representations.

## E.3 Aligning representations across threat models

Figures 32 to 36 ablate on the results of Figure 31 by varying the perturbation size of each threat model used during training. As expected, models at lower budgets tend to be better aligned. At larger budgets, even for visually similar attacks like JPEG and $\ell_2$, the aligment dips, particularly at deeper layers. This indicates that to train models jointly robust to different threat models, explicit constraints on learned representations during training may be needed.

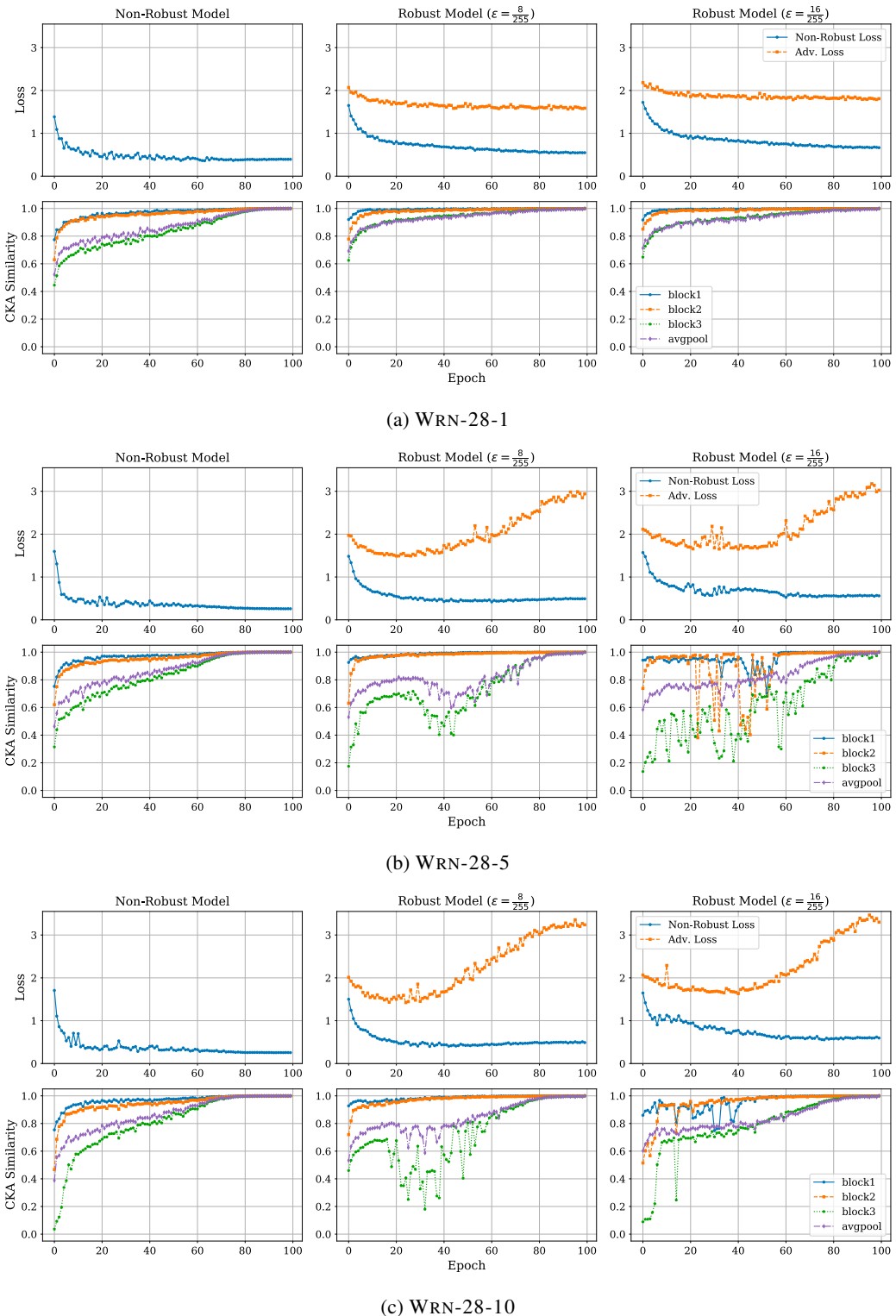

(a) WRN-28-1

(b) WRN-28-5

(c) WRN-28-10

Figure 26: **Overfitting and training instability in robust learning.** Each plot shows how benign representations of selected layers of Wide ResNet models compare to their final learned representations after each epoch of training. In larger robust models trained at $\epsilon = \frac{8}{255}$, training instability begins after the point of overfitting on adversarial data. However, when the perturbation strength is increased, there are large changes in the similarities of representations of subsequent epochs starting very early in training. High perturbation strengths also induce instability at earlier layers, which is not observed at lower strengths or in non-robust training.

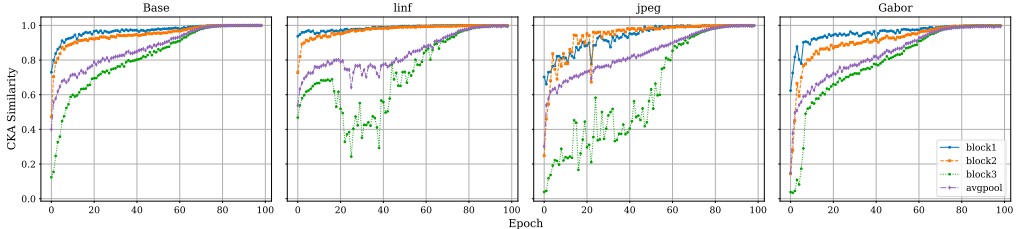

Figure 27: **Convergence of benign representations over robust training for different threat models.** Using CKA, we can observe differences in training dynamics between robust training for $\ell_\infty$, JPEG, and Gabor attacks. During JPEG-robust training, the output of block 3 appears to be even more significantly affected than in $\ell_\infty$ training, although the subsequent average pooling layer converges more smoothly. During Gabor training, the effects are more muted, possibly due to Gabor being a weaker attack.

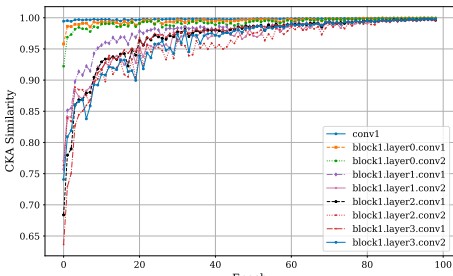

Figure 28: **Convergence of learned adversarial representations within block 1 of a robust WRN-28-5.** Zooming in on the first residual block, a clear elbow curve is present in the representations of all but the very first convolutional layers.

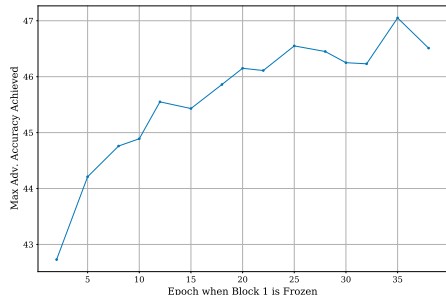

Figure 29: **Maximum adversarial validation accuracy after freezing early layers at various points during training.** Accuracies start low ( 43%) when the first block is frozen after 2 epochs of training, and steadily rise until around epoch 20, when robust accury levels off between 46 and 47%. This is similar to the pattern observed in Figure 28, in which the CKA similarity between the representations learned within block 1 at each epoch and the final representations level off around epoch 20.

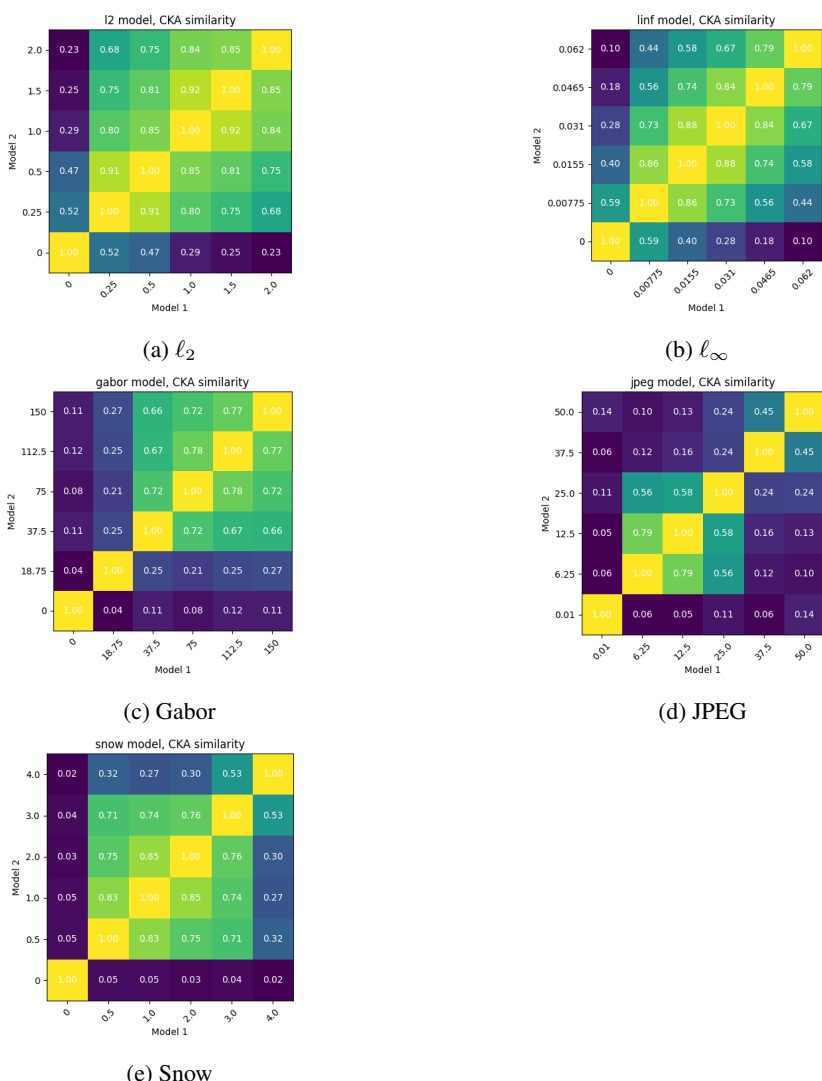

(a) $\ell_2$

(b) $\ell_\infty$

(c) Gabor

(d) JPEG

(e) Snow

Figure 30: **Adversarial training induces very different representations, even at low attack strengths.** Each plot displays the CKA similarity between activations of the final layer of WRN-28-10 feature extractors adversarially trained on different strengths of each threat model. In each plot, even the representations of the lowest strength robust model are quite dissimilar from those of the benign model, with only the $\ell_p$ bounded models having that similarity be relatively close to the lowest similarities between robust models.

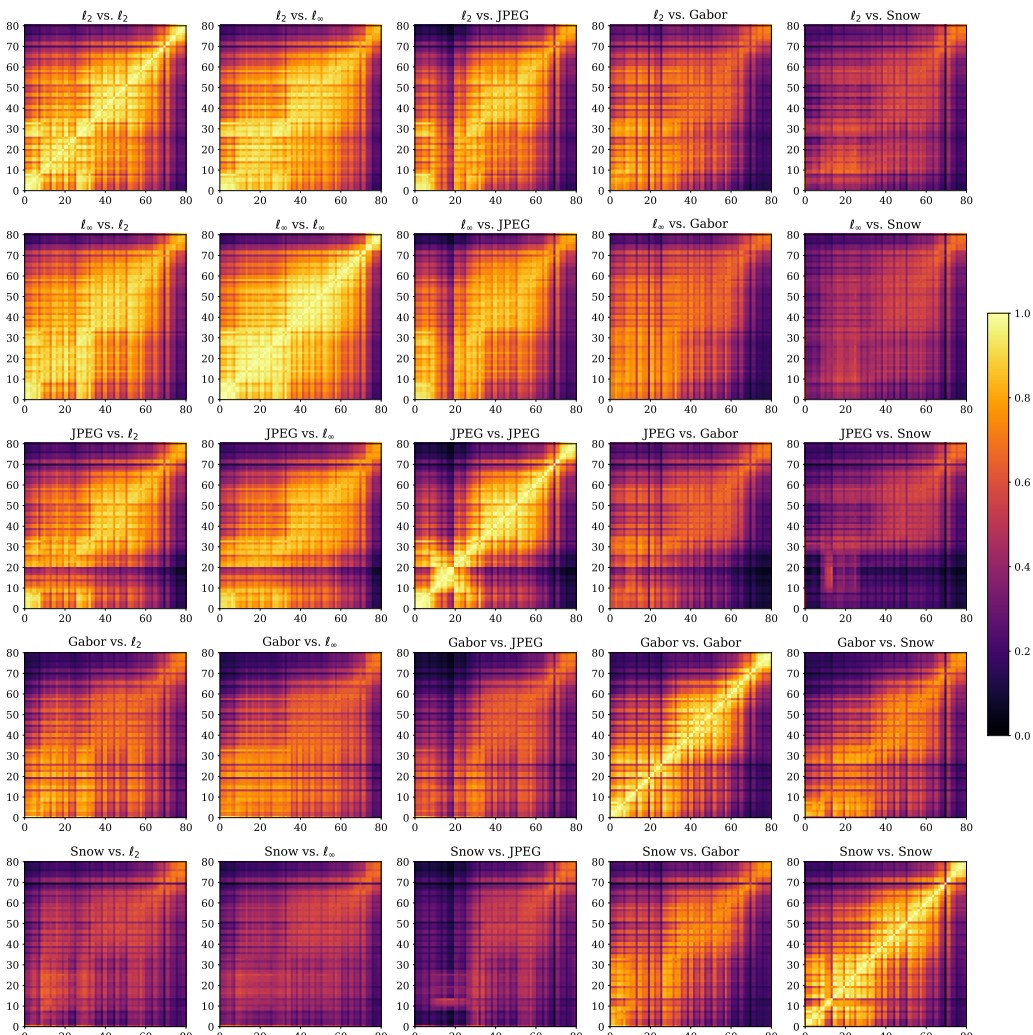

Figure 31: **Exploring relationships between threat models.** Each plot displays the layerwise similarity between benign activations of robust WRN-28-10 models trained against different threat models. As suggested by prior research [8], $\ell_p$ robust networks are shown to be highly similar. JPEG, which is $\ell_p$ bounded in the compression space, is also noticeably more similar to the $\ell_p$ bounded attacks than to the others. Interestingly, Snow and Gabor trained models are noticeably more similar to each other than to the other models, implying an unknown similarity between the classes of attack.

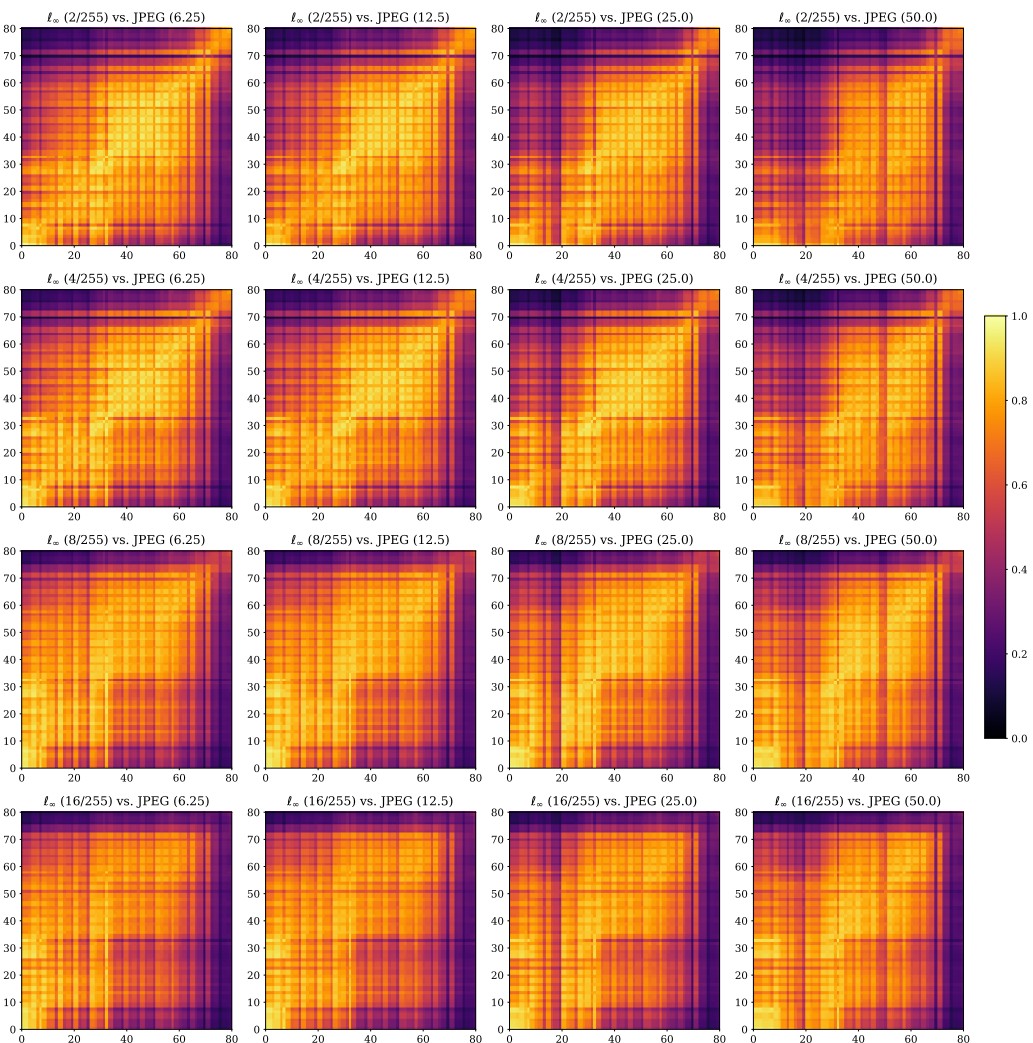

Figure 32: Comparing representations of $l_\infty$ and jpeg robust models at different attack strengths.

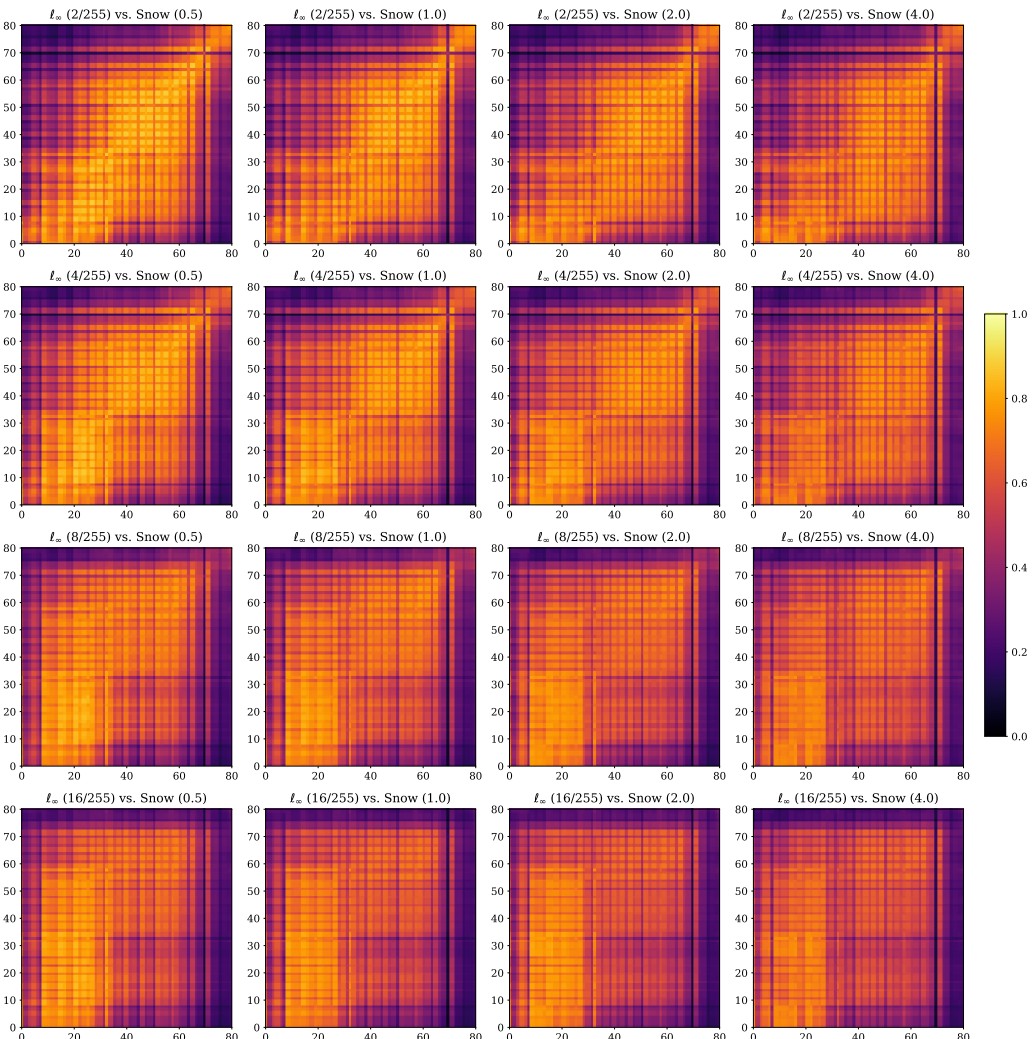

Figure 33: Comparing representations of $l_\infty$ and snow robust models at different attack strengths.

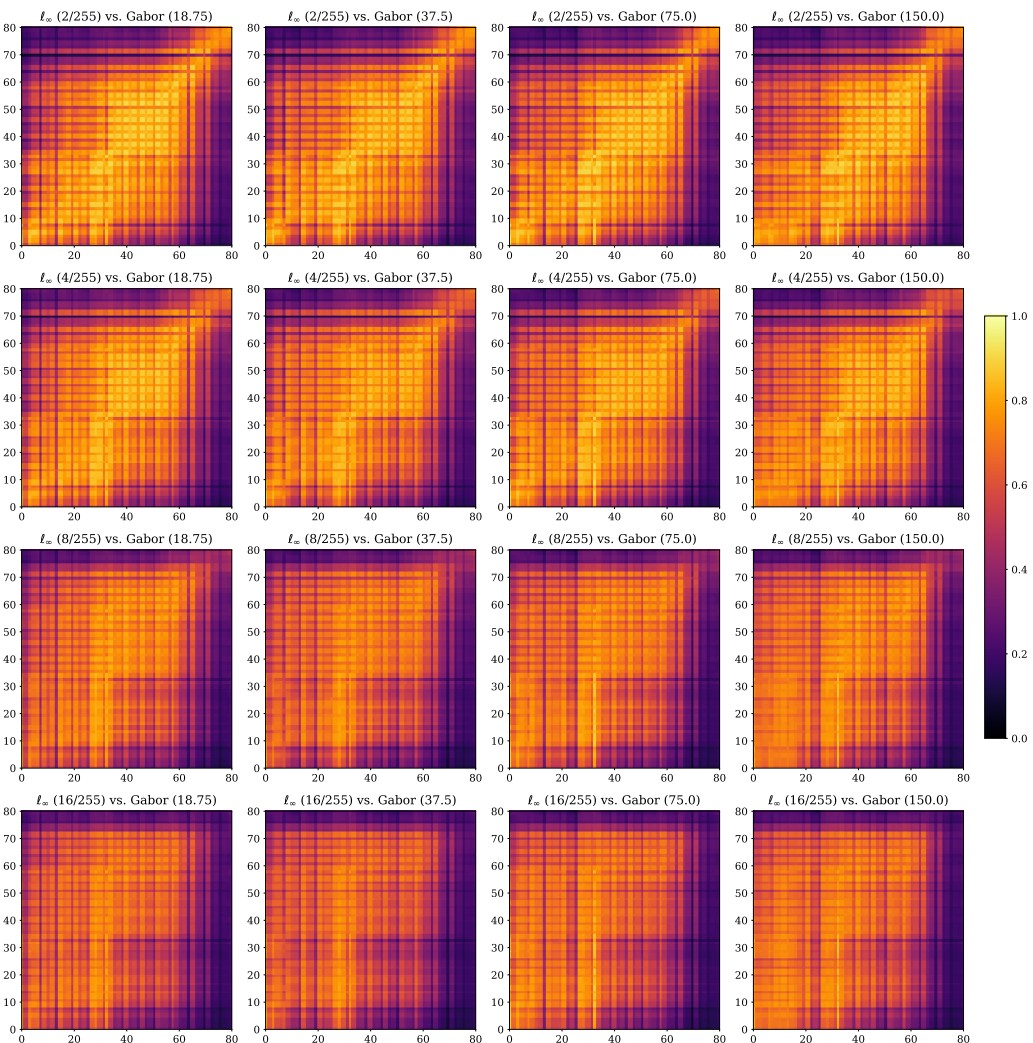

Figure 34: Comparing representations of $l_\infty$ and gabor robust models at different attack strengths.

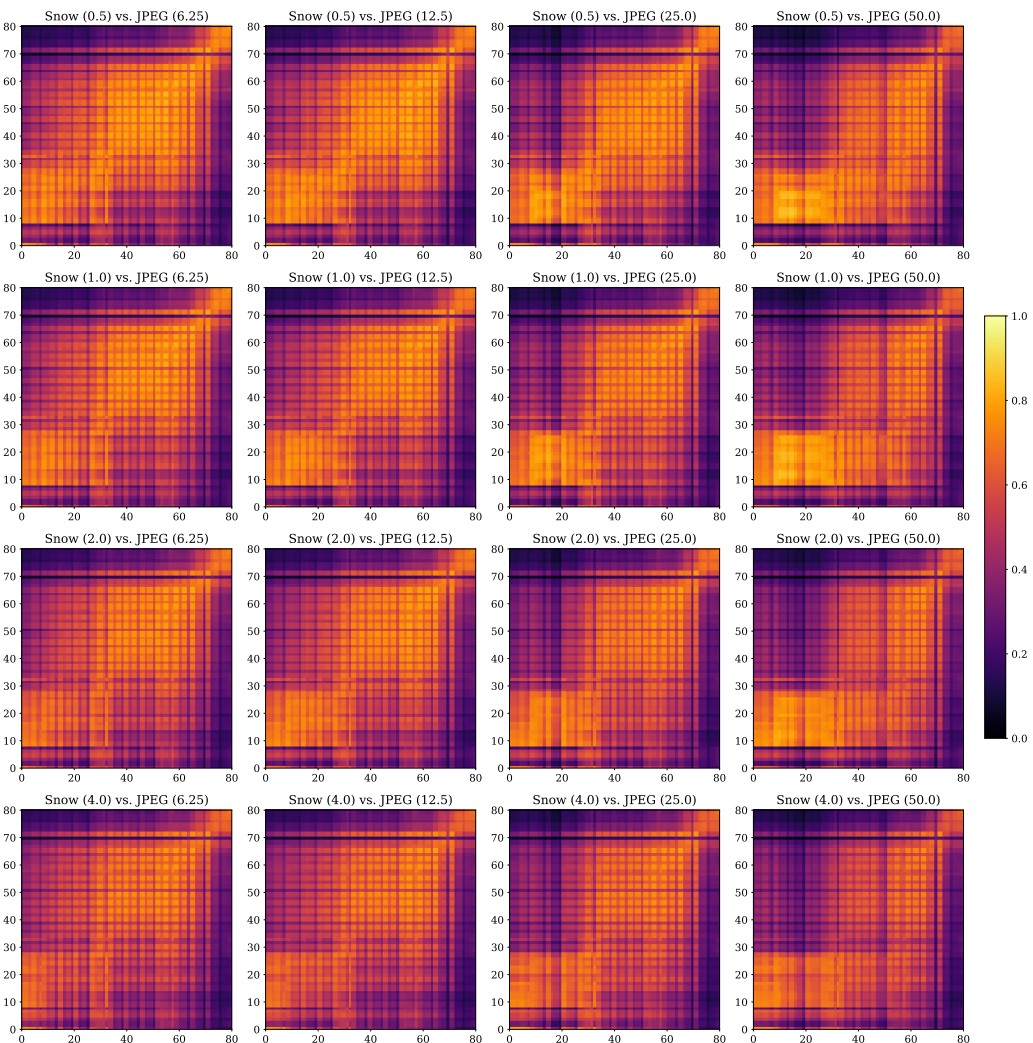

Figure 35: Comparing representations of snow and jpeg robust models at different attack strengths.

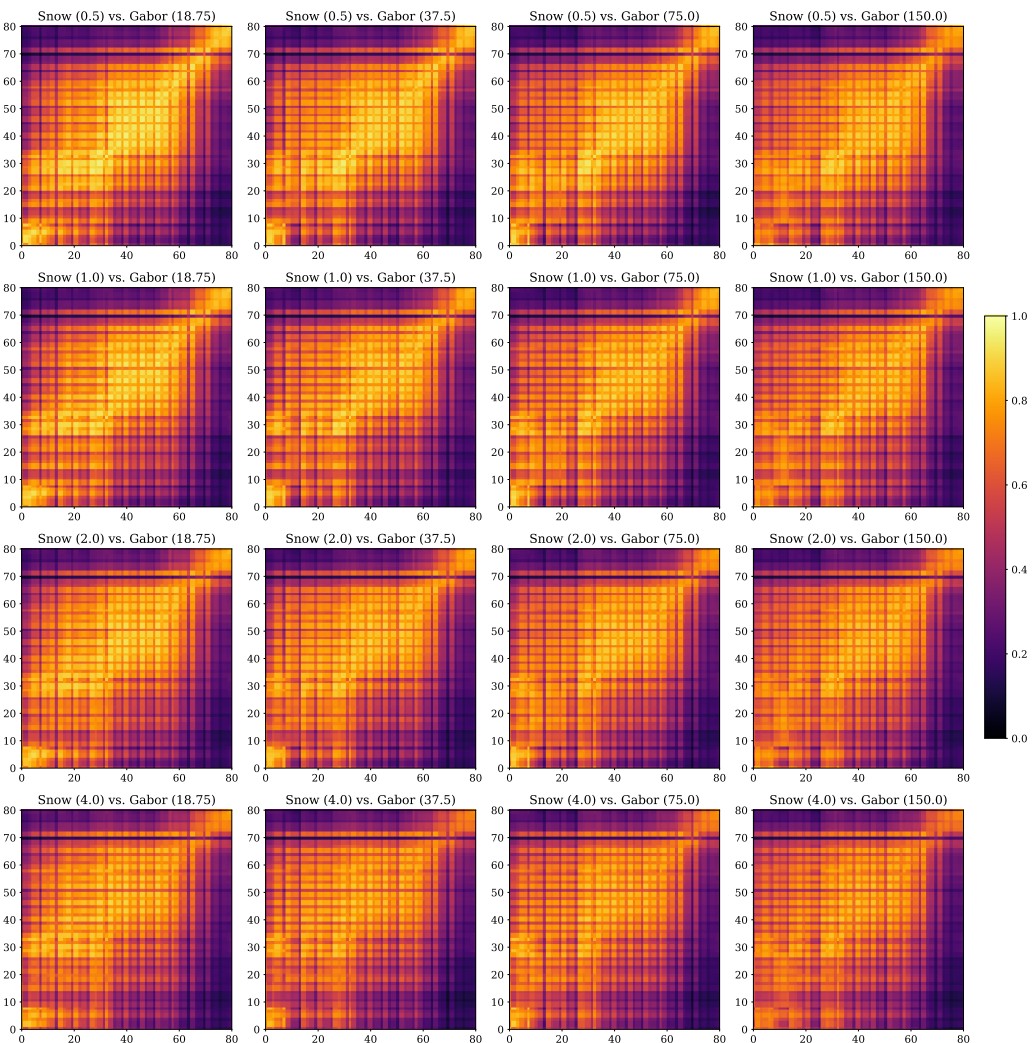

Figure 36: Comparing representations of snow and gabor robust models at different attack strengths.