# OpenReview forum: "Understanding Robust Learning through the Lens of Representation Similarities"
_NeurIPS.cc/2022/Conference — NeurIPS 2022 Accept_

### Official Review · Reviewer_9y9Q · 2022-07-11

**Rating:** 6
**Confidence:** 4
**Soundness:** 3 good
**Presentation:** 3 good
**Contribution:** 2 fair

**Summary:**

The paper contrasts representation similarities of networks trained to perform image classification with and without adversarial noise. To do so, the authors measure the similarity of representations using the Centered Kernel Alignment (CKA) metric (as well two other similarity metrics in appendix) for CIFAR-10 and two subsets of ImageNet. The authors highlight 1) networks trained with adversarial noise have layers similar to one another compared to those of standard trained networks (which have a block structure) 2) representations in early layers are unaffected by adversarial perturbations both for standard and adversarially trained networks 3) networks trained with and without adversarial perturbations have similar representations until the last 10 or so layers 4) early layers converge faster and later layers overfit to local minima. The authors also analyze the similarity of representations when other perturbations are applied JPEG, Gabor, and Snow, finding

**Questions:**

included above

**Limitations:**

Yes.

**Strengths And Weaknesses:**

This work investigates representations learned for image classification by contrasting the representation similarities of networks trained with and without adversarial noise—leading to several insights into properties of the learned representations as well as learning dynamics of modern networks. Such properties as the authors point out are not captured by aggregate performance metrics such as loss or accuracy, leading to insights about the learned representations. The paper is well-written, experiments clearly described, and motivation is well-grounded.

The experiments conducted are convincing and well-formulated. For example, multiple similarity metrics are compared, variants of the adversarial perturbations are explored, and experimental claims are well-founded.
However, the authors'claims should be sufficiently couched within the experimental settings studied: supervised image classification for ResNet-based models on CIFAR-10 and two subsets of ImageNet. For example, the work only studies ResNet-based architectures yet claims to cover “DNNs with different architectures” (line 12) and the “impact of choice of architecture” (line 37). I would expect a comparison of “different architectures” to encompass for example transformer-based architectures, MLP-architectures, etc. I suggest the authors more explicitly couch claims in the introduction, abstract, title, and conclusions within the confines of the experimental settings studied. For example
The comparison of representations for "threat models" (JPEG, Gabor, and Snow) in Section 6 was not particularly informative and seemed removed from the primary findings of the remainder of the paper.

Robust representations is a broad term. This work studies robustness to random noise in the input. Yet, the authors intermix “robustness to adversarial examples” and often plainly use the term “robust network” to describe a particular type: robustness to noise (defined via L-p bounds on the input) throughout the work. For example, robust in the context of image classification can just as well refer to robustness with respect to rendering method ImageNet-Sketch, artifacts such as blurring (ImageNet-C), adversarial examples (ImageNet-A), or even robustness to natural transformations such as pose (Alcorn et al.). I suggest the authors clarify the wording to only include the specific definitions of robustness studied here.

The insights gained from the authors’ analysis are interesting and well-described. The finding of most value, in my opinion, is that later layers overfit (matching existing work relating overfitting in later layers to spurious correlations [1]). While the analysis sheds light on differences between networks trained with and without adversarial noise as well as their learning dynamics the findings are confined to the L-infinity definition of robustness studied for supervised image classification using ResNet-based architectures (for the main claims in section 4 and 5).

[1] “Last Layer Re-Training is Sufficient for Robustness to Spurious Correlations” [https://arxiv.org/abs/2204.02937](https://arxiv.org/abs/2204.02937)

---

> ### Author Response · Authors · 2022-08-02
> **Response to Reviewer 9y9Q**
>
> We thank the reviewer for their positive appraisal of our paper and insightful comments for improving it. We address their specific questions and concerns below. The paper and supplementary have also been revised to account for all the reviewers’ feedback (see Summary of Revisions).
>
> **‘claims should be sufficiently couched within the experimental settings studied’:** We acknowledge that most of our experiments are on ResNet based models but this is largely due to their prominence in SOTA benchmarks (see top-performing models on CIFAR-10 at https://robustbench.github.io/). For the datasets we consider, MLP-based architectures do not achieve good performance, as observed with previous works that find adversarial training vision transformers challenging [1, 2].  In the meantime, we will update the text to acknowledge these limitations.
> Regardless, in the updated Section B of the Supplementary material, we have added layer-wise similarity plots for 7 additional robust training methods from the RobustBench [3] benchmark, as well as plots that use 3 additional adversarial example generation methods to generate perturbed representations. These are still ResNets, but of different widths and depths, and cover a wide range of training methods.
>
> **‘Robust representations is a broad term’:** We have edited the text where necessary to make it clear that most of our results concern robustness to adversarial examples. We would like to clarify that starting in Line 27 of the Introduction, we make it clear that the ‘robustness’ we discuss throughout the paper is worst-case robustness with respect to adversarial examples, and not random noise. In addition, the revised version of the paper now also contains baseline experiments with models robust to common corruptions (taken from the Robustbench model zoo) in Section F.5. Thus, our codebase is easily extensible to other types of corruptions and we will add further experiments to the camera-ready, if accepted.
>
> **‘later layers overfit…findings are confined to the L-infinity definition of robustness’:** We are glad the reviewer found our experiments on learning dynamics interesting and appreciate the reference to “Last Layer Re-Training is Sufficient for Robustness to Spurious Correlations”. We will add a discussion relating our results to this paper in the camera-ready, if accepted. We have also now added results on the learning dynamics of models trained to be robust to the other threat models we consider in Section F.2 of the Supplementary. These confirm our findings from the L-infinity threat model that overfitting largely happens in later layers.
>
>
> [1] Shao, Rulin, Zhouxing Shi, Jinfeng Yi, Pin-Yu Chen, and Cho-Jui Hsieh. "On the adversarial robustness of vision transformers." arXiv preprint arXiv:2103.15670 (2021).
>
> [2] Edoardo Debenedetti, “Adversarially Robust Vision Transformers”, Masters Thesis, EPFL (2022)
>
> [3] https://robustbench.github.io/

---

> > ### Comment · Reviewer_9y9Q · 2022-08-09
> > **Response accounting for author response and revisions**
> >
> > Thank you for updating the draft and the clarifying comments. My primary concern is regarding the scope of the claims made as well as the robustness definition.
> >
> > I don't believe robustness to adversarial examples is the correct term for the robustness studied here. My initial comment was intended to point out adversarial examples would encompass those in a dataset such as ImageNet-A. The variety of robustness studied in this work is in my opinion better described as robustness to adversarial perturbations (a term the authors do use in the text). The term *adversarial examples* used in the title, introduction, and text could be misleading to readers. I suggest the authors amend the use of the term to incorporate the notion of perturbations both the title, abstract, and introduction to avoid confusion in addition to clarifying the definition in the text. Claims such as "robust models exhibit X" found throughout the text are well beyond what the experiments show—they need to be appropriately phrase to account for the particular type of robustness, architecture, and limited datasets studied.
> >
> > Overall, I still find the insights gained from the authors’ analysis interesting—provided it's sufficiently clear the settings within which those claims extend.

---

> > > ### Author Response · Authors · 2022-08-09
> > > **Addressing concern regarding usage of term "adversarial examples"**
> > >
> > > We thank the reviewer for engaging with our rebuttal.
> > >
> > > In the literature, the notion of adversarial examples is commonly associated with pixel-wise perturbation-based adversarial attacks [1], hence our lack of distinction between adversarial examples generated using perturbations and other types. However, we do agree with reviewer that adversarial examples can be generated with numerous other methods, such as changes in color [2], manipulation of semantic attributes [3], patches [4], and natural adversarial examples [5]. Given the reviewer's concern on this matter, we are willing to emphasize this distinction in detail in the title, introduction, and experimental setup of the camera-ready, if accepted. We will also modify the broad term 'robust model' to 'model robust to adversarial perturbations' wherever appropriate in the text. Given the paper's experimental nature, we hoped that readers would be cognizant of the fact that the conclusions are limited to the models, datasets and threat models used, which while comprehensive, are not exhaustive. We will nonetheless articulate this point better throughout, using the extra page in the camera-ready.
> > >
> > > We hope that our rebuttal addresses most the concerns raised by the reviewer. If you find our response satisfactory, we would greatly appreciate a reconsideration of your score.
> > >
> > > 1. Akhtar, Naveed, et al. "Advances in adversarial attacks and defenses in computer vision: A survey." IEEE Access 9 (2021): 155161-155196.
> > > 2. Shamsabadi, Ali Shahin, Ricardo Sanchez-Matilla, and Andrea Cavallaro. "Colorfool: Semantic adversarial colorization." Proceedings of the IEEE/CVF Conference on Computer Vision and Pattern Recognition. 2020.
> > > 3. Hosseini, Hossein, and Radha Poovendran. "Semantic adversarial examples." Proceedings of the IEEE Conference on Computer Vision and Pattern Recognition Workshops. 2018.
> > > 4. Brown, Tom B., et al. "Adversarial patch." arXiv preprint arXiv:1712.09665 (2017).
> > > 5. Hendrycks, Dan, et al. "Natural adversarial examples." Proceedings of the IEEE/CVF Conference on Computer Vision and Pattern Recognition. 2021.

---

> > > > ### Comment · Reviewer_9y9Q · 2022-08-09
> > > > **Thank you**
> > > >
> > > > In light of the authors' willingness to sufficiently guard the reach of the claims made and clarify the wording, I've updated my score.

---

### Official Review · Reviewer_EXva · 2022-07-12

**Rating:** 7
**Confidence:** 4
**Soundness:** 3 good
**Presentation:** 3 good
**Contribution:** 3 good

**Summary:**

The authors examine the effects of adversarial robustness training on representation. Specifically, they use CKA to compare robust vs non-robust networks, benign vs. adversarial inputs, and how these comparisons change over the course of learning. They find the following results:
- Robust representations are less specialized; distant layers are more similar in robust networks, and block structure is weaker
- Early layers in robust networks are largely unaffected by adversarial examples; representations are similar for benign vs. perturbed inputs
- Deeper layers overfit during robust learning
- Models trained to be robust to different threat models have similar representations

**Questions:**

This work primarily examines adversarial (i.e. worst-case) robustness, which is one of many types of robustness, among others including average-case robustness (Hendrycks and Dietterich, Benchmarking Neural Network Robustness to Common Corruptions and Perturbations) and natural adversarial examples (Hendrycks et al.). It’s not clear how important of a problem adversarial robustness is (see Gilmer et al.’s Motivating the Rules of the Game for Adversarial Example Research). The authors examine robustness to Gabor, snow, and jpeg attacks, but the analysis and results are limited. I would encourage the researchers to more systematically examine how their findings extend to other types of robustness.

Are the results from a single instance of each model? If so, I would strongly encourage the authors to repeat their analyses in triplicate at a minimum. If this is computationally cost-prohibitive, then perhaps they could focus on the most important results.

There is relevant previous work examining the relationship between representational dimensionality and adversarial robustness (Leavitt and Morcos, Linking average- and worst-case perturbation robustness via class selectivity and dimensionality; Sanyal et al., Robustness vai Deep Low-rank Representations; Nayebi and Ganguli, Biologically inspired protection of deep networks from adversarial attacks). These papers show that low rank representations confer adversarial robustness. Can you reconcile these results with your findings that adversarial robustness is associated with greater utilization of model capacity? Additionally, most of these works (certainly Leavitt and Morcos and Sanyal et al.) regularized representational dimensionality and found that it improved robustness; it would be quite interesting to investigate whether that the inverse holds: does adversarial training reduce representational dimensionality?

The claim that “Deeper layers overfit during robust learning” seems like it could be leveraged to improve the generalization gap caused by robust learning. The simplest approach would be to simply stop training when the adversarial loss begins to rise (as in Figure 6). I assume this has drawbacks, but the authors don’t present the data (e.g. accuracy curves over training). Is this overfitting a necessary component of robust learning?

Lines 59-61: “On the other hand, the representations of benign and perturbed inputs from robust networks are indistinguishable from one another with regards to representation similarity metrics.” I think to make this claim the authors need to directly compare CKA(Benign, Perturbed) to CKA(Benign, Benign) and CKA(Perturbed, Perturbed).

The caption for Figure 2 should describe the model architecture(s) used to generate the results, as well as whether the data are benign or perturbed. There are other Figures which lack important experimental details, such as whether the data are benign or perturbed.

Comparing the CKA results in Figure 2d to the results in Figures 2a-c make it clear that the similarity between robust and non-robust networks is lower than the similarity between networks of the same type, but I think it’s important to have a negative baseline—how low is “low”? Accordingly, I think the authors should repeat the analysis (CKA between robust and non-robust networks) using sample-shuffled data (and/or some other suitable random baseline).

Showing that the block structure effect varies with the strength of adversarial training is a nice experiment and result.

The results presented in lines 174-188 (“Impact of robust training strength” and “Impact of architecture”) would be easier to interpret (and their motivation clearer) if you introduced them with Nguyen et al.’s finding that “block structure in the internal representations arises in models that are heavily overparameterized relative to the training dataset.” While you do cite their work (“Previously Nguyen et al. [28] observed that increasing network width, thus capacity, leads to emergence of block-structure in non-robust networks”), I think this is an insufficient description of their results and should be presented earlier.

“Overfitting is predominantly visible in later layers”: I would suggest a follow-up experiment to more thoroughly test this claim: freeze early layers (blocks 1 and/or 2) early in training. If early layers aren’t involved in the overfitting and training instability, then the freezing should have minimal effect on these phenomena.

Lines 278-280: “When using CKA to compare against the Snow threat model, we observe that the highest average similarity is achieved with Gabor. This represents a novel insight into these threat classes, as correspondence between the two was not previously known.” This result seems very specific and not particularly useful. I would encourage the authors to find a more general result (see my earlier comment about examining different attack types).

minor comments:

Lines 21-22: “...such as images, speech or text”. It’s a matter of personal taste, but I am a proponent of the Oxford comma: “...such as images, speech, or text”

Line 22: What does “meaningful” mean in this context? Meaningful with regards to what?

Line 187: “These results suggest that while increasing width in robust networks doesn’t lead a drastic shift in similarity of internal layer representations.” Typo?

Typo in caption of Figure 5 (...to understand similairty)


**Limitations:**

The authors devote a paragraph at the end of the discussion to the limitations of their study, which seems appropriate given the space limitations. I do, however, think the authors could be more careful and nuanced about some of their claims.

**Strengths And Weaknesses:**

Strengths: The paper follows in a well-established tradition of using representational similarity analysis to understand neural network behavior and training interventions. The analyses are well-motivated and straightforward. The results are generally presented clearly and easy to understand. The results are interesting.

Weaknesses: The paper seems somewhat limited in scope: It primarily addresses adversarial (worst-case) robustness, which is only one type of robustness. I'm not totally convinced of the utility of this work; it could at the very least do a better job situating itself within existing robustness research. It's unclear if any of the experiments were run in replicate. The figures captions could be more informative and self-contained.

Overall, I think this work could be suitable for publication if it is sufficiently revised.

---

> ### Author Response · Authors · 2022-08-03
> **Response to Reviewer EXva (Part 1 of 2)**
>
> We thank the reviewer for their detailed and constructive critique of our work and address their concerns below. The paper and supplementary have also been revised to account for all the reviewers’ feedback (see Summary of Revisions).
>
> **“It’s not clear how important of a problem adversarial robustness is …  I would encourage the researchers to more systematically examine how their findings extend to other types of robustness”**:
> The answer to this concern is a nuanced one, and we appreciate the opportunity to discuss it here. We believe that there is considerable merit in the study of adversarial robustness (see the overview of a recent workshop [2] for a nice summary), given the fundamental gaps it exposes in our understanding of the working of complex machine learning models and the fact that it represents a theoretical paradigm shift when considering issues of convergence and generalization. However, it is true that focusing too narrowly on Lp threat models, as argued by Gilmer et al., is a concern for the field. Hence, in Section 6, we have carried out experiments with other threat models as well and we agree that including other types of corruption in our analysis would strengthen our arguments. Given the flexibility of our codebase, we were able to run experiments looking at the cross-layer similarity of models robust to different types of common corruptions (Figure 28 in Section F.5 of the Appendix). We find that the effects of increased local similarity and differences between the layer-wise similarity plots for robust vs. non-robust networks are not as pronounced as they are in networks robust to worst-case adversarial attacks, implying that adversarial examples do have a particularly strong effect on representation similarity.
>
> **“Are the results from a single instance of each model? If so, I would strongly encourage the authors to repeat their analyses in triplicate at a minimum”**:
> While the reported results are from single instances, we trained multiple copies of models in most of our experiments to verify the validity of our conclusions. We have found that CKA exhibits very low standard deviation when comparing models trained with the same parameters. In our revision, we have included a new Figure 29 in Section F.6 in the Appendix that displays the standard deviation of a CKA computation (for both robust and non-robust networks). The results show a very close alignment between the three computations. In the camera-ready, if accepted, we will report standard deviations along with CKA values for our important results throughout.
>
>
> **“Previous work examining the relationship between representational dimensionality and adversarial robustness … How does adversarial training reduce representational dimensionality?”**
> We thank the reviewer for making a connection between current work and previous work on the representation dimensionality of robust networks. Our observation complements the previous work in showing that representations in adversarially robust networks do exhibit a lack of differentiation among layers, which we hypothesize is linked to greater usage of model capacity (e.g, disappearance of block structure in cross-layer similarity - Fig. 2). However, we believe that examining a causal effect of adversarial robustness on intrinsic representation dimensionality deserves an independent and rigorous evaluation of its own, including careful experimental design, which falls outside the scope of current work. We’ll be sure to discuss this interesting connection in the camera-ready version, if accepted.
>
> **“The claim that “Deeper layers overfit during robust learning” seems … Is this overfitting a necessary component of robust learning?”**
> We clarify that our objective is to understand why overfitting happens in adversarial training. Overall, it is well-known that adversarial loss at the output layer overfits [1]. Our contribution is to demonstrate that the level of overfitting differs across layers, as deeper layers overfit more during robust learning. We have also conducted experiments with early stopping, and found a large difference in the representations learned at later layers with and without early stopping, in certain cases. We can add a further detailed discussion in the camera-ready, if accepted. If the reviewer believes that our “Deeper layers overfit during robust learning” phrase is still causing a misunderstanding, we are happy to update it.
>
> **“I think to make this claim the authors need to directly compare CKA(Benign, Perturbed) to CKA(Benign, Benign) and CKA(Perturbed, Perturbed).”**
> We have included plots that display CKA (Benign, Perturbed) for multiple threat models in Figures 8, 9, and 10 in the Appendix. Plots showing CKA (Perturbed, Perturbed) for robust models were omitted due to their high visual similarity to the CKA (Benign, Benign) plots already included in the paper. If necessary, we can include a figure that directly compares the three types of plots.

---

> > ### Author Response · Authors · 2022-08-03
> > **Response to Reviewer EXva (Part 2 of 2)**
> >
> > **“[I]t’s important to have a negative baseline—how low is “low”?”**:
> >
> > We agree that context on the expected range of CKA is necessary for its use as a meaningful tool. In our experience, we have found that CKA is appropriately low when fed data that represents a random baseline. When creating a CKA similarity plot between all of the layers in a single network in which one of the activations in each comparison is the output of shuffled data, the CKA similarities for all of the comparisons are quite low (less than 0.1) and the plot appears entirely black (when visualized with the same color scale as the other plots in our paper).
> >
> > **“I would suggest a follow-up experiment to more thoroughly test this claim: freeze early layers (blocks 1 and/or 2) early in training. If early layers aren’t involved in the overfitting and training instability, then the freezing should have minimal effect on these phenomena”**:
> >
> > We thank the reviewer for suggesting this intriguing experiment, as we feel that it helps to provide a conceptual reference for the utility of our work and a practical application of the techniques we present. We have conducted experiments to test the impact of layer freezing according to the epochs indicated by our analysis. Our results indicate that a model’s accuracy increases the fastest when its internal representations are converging the fastest towards the final learned representations. To test this, we froze the first block of a WRN-28-5 at different points in the first 40 epochs of training and recorded the maximum adversarial validation accuracy achieved over 100 epochs of training. As shown in the data below, when increasing the epoch freezing occurs at, accuracy starts off low (below 43%) and steadily increases until epoch 20, after which it levels off around 46-47%. This matches the trend of the CKA similarity convergence of convolutional layers within block 1 of a WRN-28-5 network, as shown in Section F.1 of the Appendix. These results suggest that early layers are indeed not subject to the same degree of overfitting that is experienced in later layers.
> > ----------------  -------------------
> > Freezing Epoch |    Max Adv. Accuracy
> > ----------------  -------------------
> >                2                42.73
> >                5                44.21
> >                8                44.76
> >               10                44.89
> >               12                45.55
> >               15                45.43
> >               18                45.86
> >               20                46.15
> >               22                46.11
> >               25                46.55
> >               28                46.45
> >               30                46.25
> >               32                46.23
> >               35                47.05
> >               38                46.51
> >
> >
> > **“This result seems very specific and not particularly useful. I would encourage the authors to find a more general result”**:
> >
> > While we agree that this correspondence isn’t very impactful in and of itself, we believe that the primary contribution of this result is as an example of a novel discovery that can be made using CKA that is obscured when using other coarse-grained metrics like loss and accuracy. This result hints that representation similarity can be used to guide joint robust training (to multiple classes of adversarial attacks), by determining which threat models can learn similar representations and what layer. Thus, joint robust training can be guided in a more careful manner, leveraging, for example, layer freezing and weight-based regularization. We have also since conducted preliminary experiments with common corruptions (Figure 28 in Section F.5 of the Appendix), and are happy to add more results comparing representations from average- and worst-case robust models to the camera-ready, if accepted.
> >
> > ‘**insufficient description of Nguyen et al.**’ and ‘**limitations.. nuanced**’: We have updated the paper to address both of these issues.
> >
> > [1] Rice, Leslie, Eric Wong, and Zico Kolter. "Overfitting in adversarially robust deep learning." International Conference on Machine Learning. PMLR, 2020.
> > [2] https://advml-workshop.github.io/icml2021/

---

> > > ### Comment · Reviewer_EXva · 2022-08-08
> > > **Satisfactory rebuttal**
> > >
> > > Thanks for the detailed response and updates! Please remember to update the camera-ready with relevant items (discussion and references for representational dimensionality and necessity of late-layer overfitting).
> > >
> > > I have updated my scores as follows:
> > > Soundness: 2 -> 3
> > > Contribution: 2 -> 3
> > > Rating: 4 -> 7

---

### Official Review · Reviewer_HbNq · 2022-07-16

**Rating:** 6
**Confidence:** 4
**Soundness:** 2 fair
**Presentation:** 2 fair
**Contribution:** 3 good

**Summary:**

This paper uses an existing method for comparisons of intermediate neural network activations (CKA) for a comparison of robust and non-robust networks. The authors analyze the similarities in different aspects and try to deduce insights in adversarial training.

**Questions:**

- Missing related work: There has been previous work on comparing the features of robust vs. non-robust neural networks that should be cited, e.g. using feature visualizations [1].
- L25: Reference missing.
- L45f: Reference [18] should be placed behind Imagewoof and not behind Imagenet; also, a reference for ImageNet is missing.
- L93: E.g., JPEG compression doesn't seem like an additive perturbation.
- L108: If this is well known, I encourage the authors to support this sentence with (multiple) references.
- L118: A more detailed critical discussion of the possible shortcomings of these metrics and why they don't impact the results of this paper would be good.
- L123f: Why doesn't this impact your results? This definitely needs to be supported by a strong argument.
- Figure 2/3: On which datasets were these metrics calculated? On the vanilla test data or on adversarially perturbed versions of the test data? What does the figure look like for the other data type (either benign or adversarial)?
- Figure 2/3: Given that the clean accuracy goes down for adversarial training, I'm wondering whether the change in the similarity plots is really due to a special aspect of adversarial training or just due to the drop in performance, i.e. how do these plots look like for non-robust networks that are not trained until convergence but until they reach a similar test accuracy as the robust networks?
- L163f: That seems a bit like an overstatement: The block structure is still clearly visible for 2 out of the 3 datasets - it just gets weaker.
- Multiple typos/grammar mistakes that make it sometimes break the flow of the text, e.g. L172, L187, L201, L216, L221, L235.
- Figure 4: These plots should be larger - the text is hard to read.
- Figure 4: The first sentence of the caption doesn't read right/is difficult to parse.
- L185f: Did the authors observe the same behavior as described by Nguyen et al. when they tried to reproduce their observations for non-robust networks? At the moment it is difficult to confidently say that the observation reported here is because of the robust network or because of the experimental setup used by the authors.
- L213: If this is well known, I again encourage the authors to support this sentence with (multiple) references. Furthermore, this is actually not such a clear property, and there are specific attacks that just aim to create adversarial examples that transfer well between different models.
- Minor comment: The paper uses inconsistent notation - sometimes the authors say "Figure", sometimes "figure"; sometimes "Appendix", sometimes "App.".
- L252f: Where can we see both validation and training loss? The figure only shows one "loss" but doesn't say which one it is.
- L265: I wouldn't call these "adversarial perturbations" but rather use the more commonly used expression common corruptions. Especially, since at least for JPEG compression there is nothing you can optimize - so that doesn't really fit the overall adversarial framework.
- Section 6: What are the specific parameters of the JPEG, snow, and Gabor corruptions?
- L289: What is the conclusion/interpretation of the results?

[1] Leveraging Sparse Linear Layers for Debuggable Deep Networks. Eric Wong, Shibani Santurkar, Aleksander Mądry. 2021

**Limitations:**

- While the authors mentioned criticism on the similarity metric they use (CKA), they don't really explain why this doesn't apply to their analysis. This should be properly addressed.
- It is also unclear whether the results differ if a different adversarial attack (with the same thread model) was used for generating the adversarial perturbations - this should ideally also be addressed.

**Strengths And Weaknesses:**

- The paper uses an interesting & potentially insightful approach to better understand how adversarially trained networks process information and how they differ from non-robust networks.
- The attached code for this submission is well documented and looks clean, making the results more trustworthy.
- It feels like the paper spends too much time describing plots/results compared to interpreting the results and presenting hypotheses for what results mean.

---

> ### Author Response · Authors · 2022-08-03
> **Response to Reviewer HbNq**
>
> We thank the reviewer for their positive appraisal of our paper and insightful comments for improving it. We appreciate the detailed line-by-line review of the paper and have revised the paper to address the issues raised by the reviewer.
>
> We address their specific questions and concerns below. The paper and supplementary have also been revised to account for all the reviewers’ feedback (see Summary of Revisions).
>
> **Supporting references, typos, overclaims, language issues and further interpretation:** We thank the reviewer for their careful reading of the paper. Since there are a large number of small changes, we have addressed them directly in the revision.
>
> **‘critical discussion of the possible shortcomings of these metrics’:** We have added a detailed discussion of these metrics in Section A.1. of the Supplementary and updated Section 2.2 to justify the choice of CKA better. We provide more details below (as in the discussion with Reviewer 4F8X):
> * CCA and variants have some undesirable properties: The original CKA paper (Kornblith et al., 2019) points out that Canonical Correlation Analysis (CCA) and its variants are invariant to invertible linear transformations, while neural network training is not. This makes CCA fail basic sanity checks on the layer-wise similarity of networks with different random initializations (Section 6.1 of Kornblith et al.).
> * CKA is much faster: We find CKA to be 10x faster than the Procrustes metric and up to 30x faster than CCA and its variants. This speed-up allows us to get results for much larger architectures. In addition, as shown in Appendix B.1., both the CKA and Procrustes metric show a similar increase in similarity among layers for a robustly trained model, with CKA maintaining a more distinct visual structure. While it is clear that different metrics will lead to somewhat different similarity numbers, we believe our high-level conclusions will hold across valid metrics.
>
> **Impact of accuracy on similarity plots:**  We carefully design our experiments to include networks that cover a wide accuracy range. For example, we consider Wide-ResNets with width 1 to 10 (fig 3 in main paper, fig 8,9,11,12,13,15 in appendix). The size of the network increases near quadratically with width and impacts subsequent accuracy in robust training. For example, WRN-28-1 with width=1 is two orders smaller than the largest WRN-28-10 networks and achieves much lower clean and robust accuracy. By ablating across a large range of network widths, we ensured that our conclusions are agnostic to the accuracy/performance of the network. In the rebuttal, we’ve also added another ablation using models from RobustBench [2] (fig 27 in appendix) where we show that our observations are agnostic to various factors in the robust training setup.
>
> **Figure 2 and 3 for adversarially perturbed data:** From Figures 8 and 10 in the Appendix, we establish that for the robust networks we train, adversarial and benign representations are essentially identical. Thus, for space and clarity considerations, we omitted copies of Figures 2 and 3 with adversarially perturbed data. If the reviewer thinks this is a critical addition, we can add this in a straightforward manner to a future revision and the camera-ready. Further, comparing adversarially perturbed representations for benign networks is not particularly meaningful since they are not classifying the inputs correctly.
>
> **Size of Figure 4:** We will enlarge Figure 4 to reside on two rows in the camera ready, utilizing the extra page. We cannot currently enlarge it without going over the page limit.
>
> **L185f:** Yes, we could indeed reproduce similar observations to those from Nguyen et al. for non-robust networks by varying the width of the network and observing the change in block structure. These results are in Column 1 of Figure 15 in the Appendix.
>
> **L252f:** We apologize for the omission of the training loss in Figure 6. We logged the training accuracy and found that it increased steadily, but the loss was unfortunately not logged. We have the checkpoints and will add the training loss values to the plot in a further revision later this week.
>
> **‘different adversarial attack (with the same threat model)’:** We have included experiments with different attacks but for the same threat model in Section F.3 of the Appendix, observing similar results.
>
> **‘JPEG compression doesn't seem like an additive perturbation’:** We use ‘JPEG’ to refer to the JPEG attack developed in [1], not normal JPEG compression. The attack involves using JPEG compression to compress an image and then applying an adversarial perturbation to the image in the JPEG encoding space. Details of the attacks we used are included in Section A.2 of the Appendix.
>
> [1] Kang, Daniel, Yi Sun, Dan Hendrycks, Tom Brown, and Jacob Steinhardt. "Testing robustness against unforeseen adversaries." arXiv preprint arXiv:1908.08016 (2019).
>
> [2] https://robustbench.github.io/

---

> > ### Comment · Reviewer_HbNq · 2022-08-07
> > **Response to Rebuttal**
> >
> > Dear authors,
> >
> > Thanks for your detailed response and for integrating some of the suggested changes in your manunscript! I know believe that your paper is a solid contribution to the field and fits into NeurIPS. However, as the immediate impact of this paper is a bit unclear to me, I tend to keep my current overall rating as its description ("Technically solid, moderate-to-high impact paper [...]") best describes your work.

---

### Official Review · Reviewer_4F8X · 2022-07-16

**Rating:** 4
**Confidence:** 4
**Soundness:** 2 fair
**Presentation:** 3 good
**Contribution:** 2 fair

**Summary:**

This paper presents a probing analysis on clean (non-robust) vs. adversarially trained robust models. The paper's novelty is questionable and the insights gained are also sort of obvious, e.g., that the representational differences between inputs increase as one goes deeper down the layers. While this is true (and has been observed by the authors in their experiments), the authors didn't make an attempt to analyze whether two inputs from the same class are do have similar representations, which is a good thing. How does this compare between robust and non-robust models?

The paper introduces a lot of defence mechanisms (mainly centred around the min-max idea). However, it only employs PGD asdversarial training for obtaining a "robust" model. What about other approaches that provide defence mechanisms against potential attacks, e.g. the references 23, 32, 43 etc. that the authors themselves cite?



**Questions:**

How do the analysis from the observational differences between robust and non-robust networks can actually be used in practice? Can we use these insights to guide us towards constructing more robust models?

**Ethics Review Area:**

["I don’t know"]

**Limitations:**

Limitations are not properly explained. The expression "fundamental disjunct between aggregate properties and layer-wise representation similarity metrics..." is rather vague.

**Strengths And Weaknesses:**

Strengths:

1. Good analysis work on robust vs. non-robust networks.
2. Uses CKA to measure representational similarities.

Weaknesses:

1. Only one version of robust network considered - those trained with PGD based adversarial examples.
2. The rational of using CKA is not appropriately justified.
3. Some analysis could/should have been at a more detailed level, like one would still want the differences between different classes to be high and similarties between identical classes to be high. What observations can we make regarding this expected behavior in a robust and a non-robust network?

---

> ### Author Response · Authors · 2022-08-02
> **Response to Reviewer 4F8X (1/2)**
>
> We thank the reviewer for their considered critique of the paper and address their concerns and comments below. The paper and supplementary have also been revised to account for all the reviewers’ feedback (see Summary of Revisions).
>
> ‘**analyze whether two inputs from the same class have similar representations**’: This is a very interesting question that we have already investigated in the paper. In Section B.5. of the Supplementary, we derive a small extension of the CKA metric needed to answer this question. We split up the CKA computation into an intra-class and inter-class component. We summarize our findings here and request the reviewer to go through our detailed results in B.5. We find that while there are far fewer intra-class terms in the CKA computation (for balanced label sets), they have a far higher contribution than the inter-class terms. Thus, we do observe that for both robust and non-robust networks, representations from the same class have a high degree of similarity, while those from different classes are far less similar. In addition, we tested whether inter-class similarities were lower for robust networks than non-robust ones, to understand whether this could be contributing to their lower accuracy. This was true for the CIFAR-10 and Imagenette datasets, but not for Imagewoof.
>
> ‘**only one version of robust network**’: We respectfully disagree with this comment from the reviewer, since the epoch-wise layer similarity plots in Figure 6 of the main paper and Figure 5 in the supplementary both consider another state-of-the-art training method, TRADES. This method uses a weighted sum of losses on benign and robust data to train robust models. Nevertheless, in the updated Section F of the Supplementary material, we have added layer-wise similarity plots for 7 additional robust training methods from the RobustBench benchmark, as well as plots that use 2 additional adversarial example generation methods to generate perturbed representations.
>
> ‘**rationale of using CKA**’: We thank the reviewer for this clarifying question. We have added further details justifying our choice of CKA in Section 2.2, which we summarize here:
> CCA and variants have some undesirable properties: The original CKA paper (Kornblith et al., 2019) points out that Canonical Correlation Analysis (CCA) and its variants are invariant to invertible linear transformations, while neural network training is not. This makes CCA fail basic sanity checks on the layer-wise similarity of networks with different random initializations (Section 6.1 of Kornblith et al.).
> CKA is much faster: We find CKA to be 10x faster than the Procrustes metric and up to 30x faster than CCA and its variants. This speed-up allows us to get results for much larger architectures. In addition, as shown in Appendix B.1., both the CKA and Procrustes metric show a similar increase in similarity among layers for a robustly trained model, with CKA maintaining a more distinct visual structure. While it is clear that different metrics will lead to somewhat different similarity numbers, we believe our high-level conclusions will hold across valid metrics.

---

> ### Author Response · Authors · 2022-08-02
> **Response to Reviewer 4F8X (2/2)**
>
> ‘**analysis … actually be used in practice**’: This is a great question (also asked by Reviewer mj83) and one which we hoped would arise from our analysis of robust representations. We will just note that the actual training of more robust networks is somewhat tangential to the goals of this paper, which were mainly to explore properties of robust representations from current training methods.
>
> Nevertheless, we envision **3 key ways** in which our results could be used to train better robust networks in a more efficient manner:
> - *Staggered freezing of layers during training*: Our results from Section 5 indicate that early layers do not need to be updated post a few epochs of training, since their learned representations do not change much during training. As pointed out by Reviewer EXva, we have conducted experiments to test the impact of layer freezing according to the epochs indicated by our analysis. Our results indicate that a model’s accuracy increases the fastest when its internal representations are converging the fastest towards the final learned representations. To test this, we froze the first block of a WRN-28-5 at different points in the first 40 epochs of training and recorded the maximum adversarial validation accuracy achieved over 100 epochs of training. As shown in the data below, when increasing the epoch freezing occurs at, accuracy starts off low (below 43%) and steadily increases until epoch 20, after which it levels off around 46-47%. This matches the trend of the CKA similarity convergence of convolutional layers within block 1 of a WRN-28-5 network, as shown in Section F.1 of the Appendix. These results suggest that knowledge of a network’s training dynamics derived from CKA analysis can be used to increase the efficiency of training through the freezing of early layers.
>
> Freezing Epoch  |  Max Adv. Accuracy
> ----------------  -------------------
>                2                42.73
>                5                44.21
>                8                44.76
>               10                44.89
>               12                45.55
>               15                45.43
>               18                45.86
>               20                46.15
>               22                46.11
>               25                46.55
>               28                46.45
>               30                46.25
>               32                46.23
>               35                47.05
>               38                46.51
>
> *Increasing layer-wise differentiation during training*: Our results show there is a much greater degree of local similarity among learned representations for robust networks when compared to benign ones. This similarity also increases when the training budget is increased. We suspect this lack of layer-wise differentiation may be part of the reason why robust networks do not achieve high accuracy on clean data. Using regularization methods that promote increased layer-wise differentiation during robust training may alleviate this issue, and is a compelling and immediate experiment for future work.
>
> *Choosing threat models for joint robust training*: Past work on the training of models jointly robust to multiple types of attacks has largely focused on different types of Lp perturbations. We posit that our analysis of the similarity of representations obtained from different threat models can be utilized to determine against which sets of threat models joint robustness is possible, and if the model has sufficient capacity for that purpose. Further, the layer-wise analysis can be used to add appropriate regularizers to ensure convergence, an issue exacerbated by the presence of multiple types of adversarial examples.
> We will add a summary of this discussion to the camera-ready as well, if accepted. The results on layer freezing will be added to Section 5, and the other future work to Section 7.
>
> ‘**Limitations are not properly explained**’: We apologize for the lack of a more detailed limitations section and have revised the paper to clarify its limitations. In particular, we acknowledge limitations with respect to the dependence of the results on the particular metric used, the sometimes tenuous link between properties such as accuracy and layer-wise structure, and that detailed experiments on improved robust training needed to be pushed to future work for space and time considerations.

---

> ### Author Response · Authors · 2022-08-09
> **Any feedback on rebuttal and revision?**
>
> We hope that our new experimental results and detailed comments address all of your concerns. If you find our rebuttal and revision satisfactory, we would greatly appreciate a reconsideration of your score for the paper.

---

### Official Review · Reviewer_mj83 · 2022-07-19

**Rating:** 6
**Confidence:** 2
**Soundness:** 4 excellent
**Presentation:** 3 good
**Contribution:** 3 good

**Summary:**

This paper studies the robustness of deep neural networks based on the perspective of representation similarity.  Such a perspective provides an interesting direction to delve deeper into the properties of robust representation learning. The authors make several novel discoveries on "salient pitfalls" in robust networks. According to the observations, the author introduces several ways to design and train better robust networks.

**Questions:**

1.  Is there possible to involve more experimental results on how the proposed ways for better robust learning could improve upon the previous methods?

**Ethics Review Area:**

["I don’t know"]

**Limitations:**

yes.

**Strengths And Weaknesses:**

Strength:
1. Analyzing the properties of robustness from the representation similarities is intuitive and has been explored by some previous works. However, the paper provided a very systematic study and provided lots of interesting discoveries.

2. The paper is well written. The analysis and discussion are conducted in a very logical way.

3. There are extensive experiments conducted to demonstrate the conclusion and observation. I believe the results are solid.


Weakness:
1.  I appreciate the efforts in providing different views/frameworks for an important research problem. While I would also like to find more practical effects of the proposed analysis and observations.

---

> ### Author Response · Authors · 2022-08-02
> **Response to Reviewer mj83**
>
> We thank the reviewer for their positive appraisal of our paper and interesting questions. We address their specific concerns and comments below. The paper and supplementary have also been revised to account for all the reviewers’ feedback (see Summary of Revisions).
>
> ‘**practical effects of the proposed analysis**': This is a great question (also asked by Reviewer 4F8X) and one which we hoped would arise from our analysis of robust representations. We will just note that the actual training of more robust networks is somewhat tangential to the goals of this paper, which were mainly to explore properties of robust representations from current training methods.
>
> Nevertheless, we envision **3 key ways** in which our results could be used to train better robust networks in a more efficient manner:
> - *Staggered freezing of layers during training*: Our results from Section 5 indicate that early layers do not need to be updated post a few epochs of training, since their learned representations do not change much during training. As pointed out by Reviewer EXva, we have conducted experiments to test the impact of layer freezing according to the epochs indicated by our analysis. Our results indicate that a model’s accuracy increases the fastest when its internal representations are converging the fastest towards the final learned representations. To test this, we froze the first block of a WRN-28-5 at different points in the first 40 epochs of training and recorded the maximum adversarial validation accuracy achieved over 100 epochs of training. As shown in the data below, when increasing the epoch freezing occurs at, accuracy starts off low (below 43%) and steadily increases until epoch 20, after which it levels off around 46-47%. This matches the trend of the CKA similarity convergence of convolutional layers within block 1 of a WRN-28-5 network, as shown in Section F.1 of the Appendix. These results suggest that knowledge of a network’s training dynamics derived from CKA analysis can be used to increase the efficiency of training through the freezing of early layers.
>
> Freezing Epoch  |  Max Adv. Accuracy
> ----------------  -------------------
>                2                42.73
>                5                44.21
>                8                44.76
>               10                44.89
>               12                45.55
>               15                45.43
>               18                45.86
>               20                46.15
>               22                46.11
>               25                46.55
>               28                46.45
>               30                46.25
>               32                46.23
>               35                47.05
>               38                46.51
> - *Increasing layer-wise differentiation during training*: Our results show there is a much greater degree of local similarity among learned representations for robust networks when compared to benign ones. This similarity also increases when the training budget is increased. We suspect this lack of layer-wise differentiation may be part of the reason why robust networks do not achieve high accuracy on clean data. Using regularization methods that promote increased layer-wise differentiation during robust training may alleviate this issue, and is a compelling and immediate experiment for future work.
> - *Choosing threat models for joint robust training*: Past work on the training of models jointly robust to multiple types of attacks has largely focused on different types of Lp perturbations. We posit that our analysis of the similarity of representations obtained from different threat models can be utilized to determine against which sets of threat models joint robustness is possible, and if the model has sufficient capacity for that purpose. Further, the layer-wise analysis can be used to add appropriate regularizers to ensure convergence, an issue exacerbated by the presence of multiple types of adversarial examples.
>
> We will add a summary of this discussion to the camera-ready as well, if accepted. The results on layer freezing will be added to Section 5, and the other future work to Section 7.

---

> > ### Comment · Reviewer_mj83 · 2022-08-08
> > **Reply to Rebuttal**
> >
> > Thank the authors for their detailed reply. The involved results and discussions make the picture clear for me. I tend to keep my score and vote for acceptance.

---

### Author Response · Authors · 2022-08-02
**Summary of responses and first revision to paper**

We thank the reviewers for their thoughtful and constructive engagement with the paper. As reviewers ourselves, we greatly appreciate the reviewers’ efforts at providing thorough and insightful commentary on the paper.

We are glad that multiple reviewers found the paper to be well-grounded and motivated (**EXva, 9y9Q**), with an interesting approach (**HbNq**). We appreciate the acknowledgement of systematic and well-formulated experiments (**mj83, 9y9Q**) leading to interesting results (**mj83, EXva, 9y9Q**) and well-described insights (**9y9Q**). Keeping in mind the importance of open-source code for reproducible research, we are happy that our code was deemed to be ‘clean and well-documented’ (**HbNq**).

We have revised both the main body and supplementary material in accordance with suggestions from the reviewers. Due to space limitations on the main body for the revision, all **new experiments and results are in the self-contained Section F in the Supplementary Material for ease of reading**. We will move results and explanations to the main body for the camera-ready (utilizing the extra page) as appropriate, if accepted. We list the key revisions to the paper and additional experiments in the Supplementary (tagged by reviewer id) below:
1. Corrected typos and tightened captions throughout
2. Added clarifications to the Introduction regarding the scope of architectures and types of robustness considered (**EXVa**, **9y9Q**)
3. Justification of CKA in Section 2.2 (**4F8X**, **HbNq**)
4. Updated limitations in Section 7 (**4F8X**)
5. Layer freezing experiment details and results in Section F.1 of the Supplementary (**mj83**, **4F8X**, **EXVa**)
6. Threat model training plots in Section F.2 of the Supplementary (**9y9Q**)
7. Layerwise similarity plots for further attack types in Section F.3 of the Supplementary (**HbNq**)
8. Layerwise similarity plots for more training methods and architectures in Section F.4 of the Supplementary (**4F8X**,**9y9Q**)
9. Layerwise similarity plots for common corruptions in Section F.5 of the Supplementary (**9y9Q**)
10. Baseline experiments in triplicate (**EXva**)

---

### Meta-Review · Area_Chair_8RSs · 2022-08-27

**Recommendation:** Accept
**Confidence:** Certain

**Metareview:**

The authors study representations obtained from image classifiers and contrast the classic training with adversarial training, so-called non-robust and robust networks, respectively. The authors primarily use the CKA metric on CIFAR10 and subsets of ImageNet2012 provide several novel insights on "salient pitfalls" in robust networks which suggest that robust representations are less specialized with a weaker block structure, early layers in robust networks are largely unaffected by adversarial examples as the representations seem similar for benign vs. perturbed inputs, deeper layers overfit during robust learning, and that models trained to be robust to different threat models have similar representations.

The reviewers agreed that these contributions are interesting to the larger community and that the presentation of the results is clear and straightforward. The main issues raised by the reviewers were carefully addressed in the rebuttal. Please update the manuscript as discussed.

**Award:**

No

---

### Decision · Program_Chairs · 2022-09-14

Accept